# A Flexible Nadaraya-Watson Head Can Offer Explainable and Calibrated Classification

**Alan Q. Wang**                                                                 *aw847@cornell.edu*
*School of Electrical and Computer Engineering*
*Cornell University*

**Mert R. Sabuncu**                                                          *msabuncu@cornell.edu*
*School of Electrical and Computer Engineering*
*Cornell University*

**Reviewed on OpenReview:** *https://openreview.net/forum?id=iEq6lhG4O3*

## Abstract

In this paper, we empirically analyze a simple, non-learnable, and nonparametric Nadaraya-Watson (NW) prediction head that can be used with any neural network architecture. In the NW head, the prediction is a weighted average of labels from a support set. The weights are computed from distances between the query feature and support features. This is in contrast to the dominant approach of using a learnable classification head (e.g., a fully-connected layer) on the features, which can be challenging to interpret and can yield poorly calibrated predictions. Our empirical results on an array of computer vision tasks demonstrate that the NW head can yield better calibration with comparable accuracy compared to its parametric counterpart, particularly in data-limited settings. To further increase inference-time efficiency, we propose a simple approach that involves a clustering step run on the training set to create a relatively small distilled support set. Furthermore, we explore two means of interpretability/explainability that fall naturally from the NW head. The first is the label weights, and the second is our novel concept of the "support influence function," which is an easy-to-compute metric that quantifies the influence of a support element on the prediction for a given query. As we demonstrate in our experiments, the influence function can allow the user to debug a trained model. We believe that the NW head is a flexible, interpretable, and highly useful building block that can be used in a range of applications.

## 1 Introduction

Many state-of-the-art classification models are parametric deep neural networks, which can be viewed as a combination of a deep feature extractor followed by one or more fully-connected (FC) classification layers (He et al., 2015; Huang et al., 2016; Dosovitskiy et al., 2020). Two common problems with parametric deep learning models are that they can be hard to interpret (Li et al., 2021) and often suffer from poor calibration (Guo et al., 2017).

Nonparametric (and, similarly, attention-based) modeling strategies have been explored extensively in deep learning (Wilson et al., 2015; Papernot & McDaniel, 2018; Chen et al., 2018; Kossen et al., 2021; Laenen & Bertinetto, 2020; Iscen et al., 2022; Frosst et al., 2019). In this approach, the prediction for a given test input (which we will also refer to as a query) is computed as an explicit function of training datapoints. These models can be more interpretable, since the dependence on other datapoints gives an indication about what is driving the prediction Hanawa et al. (2021); Taesiri et al. (2022). Yet, they can suffer from significant computational overhead, can take a hit in performance versus parametric approaches, and/or require complex approximate inference schemes (Bui et al., 2016; Salimbeni & Deisenroth, 2017).

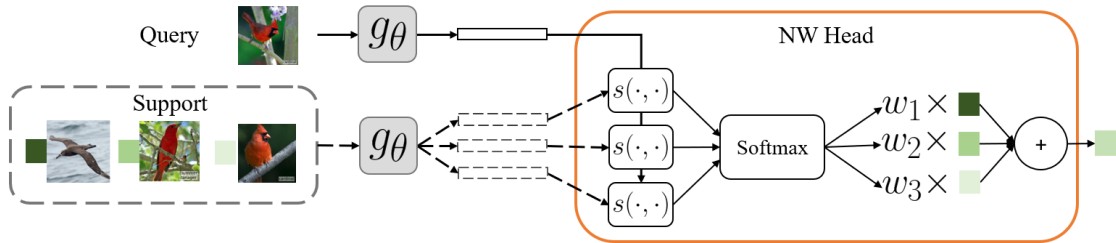

Figure 1: The NW head. Query image and support images are fed through a feature extractor $g_\theta$. Pairwise similarities $s(\cdot, \cdot)$ (e.g. negative Euclidean distance) are computed between the query and each support feature. These similarities are normalized and used as weights for computing the weighted sum of corresponding support labels.

In this work, we present an analysis of a Nadaraya-Watson (NW) head, a simple, non-learnable, and non-parametric prediction layer based on the Nadaraya-Watson model (Nadaraya, 1964; Watson, 1964; Bishop, 2006). For a given query, a pre-defined similarity is computed between the query feature and each feature in a "support set", which can be made up of samples from the training data. These similarities are normalized to weights, which are in turn used to compute a weighted average of the labels in the support set used as the final prediction.

As we discuss in this paper, the choice of the support set for the NW head affords the user an additional degree of freedom in implementation. We demonstrate several ways of constructing the support set, including a simple support set distillation approach based on clustering, and discuss the relative advantages and trade-offs. In our experiments, we show that the NW head can achieve comparable (and sometimes better) performance than the FC head on a variety of computer vision datasets. In particular, we find that the NW head exhibits better calibrated predictions. Improvements are particularly pronounced in small to medium-sized fine-grained classification tasks, where there is little sacrifice to computation time.

In addition, we explore two means of interpretability/explainability which fall naturally from the NW head. The first is interpreting the support set weights, which directly correspond to the degree of contribution a specific training datapoint has on the prediction. The second is our novel concept of "support influence", which highlights helpful and harmful support examples and can be used as a diagnostic tool to understand and explain model behavior. In our experiments, we demonstrate the utility of these methods for purposes of interpreting, explaining, and potentially intervening in model predictions.

## 2 Related Work

### 2.1 Nonparametric Deep Learning and Attention

Nonparametric models in deep learning have received much attention in previous work. Deep Gaussian Processes (Damianou & Lawrence, 2013), Deep Kernel Learning (Wilson et al., 2015), and Neural Processes (Kossen et al., 2021) build upon Gaussian Processes and extend them to representation learning. Other works have generalized $k$-nearest neighbors (Papernot & McDaniel, 2018; Taesiri et al., 2022), decision trees (Zhang et al., 2018), density estimation (Fakoor et al., 2020), and more general kernel-based methods (Nguyen et al., 2021; Zhang et al., 2021; Ghosh & Nag, 2001) to deep networks and have explored the interpretability that these frameworks provide.

Closely-related but orthogonal to nonparametric models are attention-based models, most notably self-attention mechanisms popularized in Transformer-based architectures in natural language processing (Vaswani et al., 2017) and, more recently, computer vision (Dosovitskiy et al., 2020; Jaegle et al., 2021; Parmar et al., 2018). Retrieval models in NLP learn to look up relevant training instances for prediction (Sachan et al., 2021; Borgeaud et al., 2021) in a nonparametric or semiparametric manner. Many works have recognized the inherent interpretable nature of attention and have leveraged it in vision (Chen

et al., 2018; Guan et al., 2018; Wang et al., 2017; Xu et al., 2015). More recently, nonparametric transformers (NPT) (Kossen et al., 2021) applied attention in a nonparametric setting. NPT leverages self- and cross-attention between both attributes and datapoints, which are stacked up successively in an alternating fashion to yield a deep, non-linear architecture.

Building upon and in contrast to these prior works, we present a comparatively simple nonparametric head which is computationally-efficient and performant, and which additionally lends itself naturally to being interpretable.

## 2.2 Metric Learning

Metric learning seeks to quantify the similarity between datapoints for use in downstream tasks (e.g., unsupervised pretraining (Wu et al., 2018) or retrieval (Chen et al., 2021)). Of particular relevance to our method is metric learning in the context of low-shot classification (Snell et al., 2017; Vinyals et al., 2016; Sung et al., 2017; Santoro et al., 2016). In particular, Matching Networks (Vinyals et al., 2016) for low-shot and meta-learning are most similar to our model in the sense that it uses an NW-style model to compute predictions, where both the query image and support set are composed of examples from *unseen* classes. In contrast, we analyze the NW head from a nonparametric perspective in the typical, many-shot image classification setting.

## 2.3 Influence Functions

Following classical work by Cook & Weisberg (1982), Koh & Liang (2017) propose the influence function as a means of explaining black-box neural network predictions. Let $h$ denote a function parameterized by $\theta$. $\mathcal{D} = \{z_i : (x_i, y_i)\}_{i=1}^n$ is a set of training examples and $L(h_\theta(z))$ indicates the loss for a training example $z$. $\theta^*$ is the set of optimal parameters which minimizes the empirical risk. Similarly, the parameter set associated with up-weighting a training example $z$ by $\epsilon$ is found by solving:

$$\arg\min_\theta \frac{1}{n} \sum_{i=1}^n L(h_\theta(z_i)) + \epsilon L(h_\theta(z)). \tag{1}$$

Koh & Liang (2017) define influence as the change in the loss value for a particular test point $z_t$ when a training point $z$ is up-weighted. Using a first-order Taylor expansion and the chain rule, influence can be approximated by a closed-form expression:

$$\mathcal{I}(z_t, z) = -\nabla L(h_{\theta^*}(z_t))^T H_{\theta^*}^{-1} \nabla L(h_{\theta^*}(z)), \tag{2}$$

where $H_{\theta^*}$ represents the Hessian with respect to $\theta^*$. $\mathcal{I}(z_t, z)/n$ is approximately the change in the loss for the test-sample $z_t$ when a training sample $z$ is removed from the training set. However, this result is based on the assumption that the underlying loss function is strictly convex in $\theta$ and that the Hessian matrix is positive-definite. Follow-up work has questioned the efficacy of influence functions in practical scenarios where these assumptions are violated (Basu et al., 2020).

In this work, we define the support influence function which follows naturally from the NW head; it computes the change in the loss directly on the prediction vector itself (and not indirectly through the model parameters $\theta$). Furthermore, it is easy-to-compute, requires no approximations, and does not require any assumptions on convexity or differentiability.

# 3 Methods

## 3.1 Nadaraya-Watson Head

Consider a "support set" of $N_s$ examples and their associated labels, $\mathcal{S} = \{z_i : (x_i, y_i)\}_{i=1}^{N_s}$. This support set can be a subset, or the whole, of the training dataset. In this paper, we will focus on the image classification setting, where $x$'s are images and $y$'s are integer class labels from a label set $\mathcal{C} = \{1, ..., C\}$. We will use $\vec{y}$ to

denote the one-hot encoded version of $y$. We note that the NW head can be used with any input data type, not just images, and can be readily extended to the regression setting.

For a query image $x$, the Nadaraya-Watson prediction is computed as the weighted sum of support labels $\vec{y}_i$, where the weights quantify the similarity of the query image with each support image $x_i$:

$$f(x, \mathcal{S}) = \sum_{i=1}^{N_s} w(x, x_i) \vec{y}_i. \tag{3}$$

The weights can be thought of as attention kernels (Vinyals et al., 2016) and are similar to attention mechanisms in deep learning (Vaswani et al., 2017; Dosovitskiy et al., 2020; Kossen et al., 2021). Classically, these weights are defined as a function of handcrafted kernel functions $\kappa$:

$$w(x, x_i) = \frac{\kappa(x, x_i)}{\sum_{j=1}^{N_s} \kappa(x, x_j)}, \tag{4}$$

where $\kappa(\cdot, \cdot)$ quantifies the non-negative similarity between its two arguments (Bishop, 2006). Higher similarity values correspond to more similar pairs of datapoints.

In the NW head, we define the weights as:

$$w_\theta(x, x_i) = \frac{\exp\left\{-\|g_\theta(x) - g_\theta(x_i)\|_2/\tau\right\}}{\sum_{j=1}^{N_s} \exp\left\{-\|(g_\theta(x) - g_\theta(x_j))\|_2/\tau\right\}}. \tag{5}$$

Here, $g_\theta$ is a feature extractor with parameters $\theta$, which we implement as a neural network. $\tau$ is a temperature hyperparameter; unless otherwise stated, we set $\tau = 1$. $\|\cdot\|_2$ denotes Euclidean distance. Since $\sum_{i=1}^{N_s} w_\theta(x, x_i) = 1$ and $\vec{y}_i$'s are one-hot encoded vectors, the output $f$ of the estimator can be interpreted as a conditional probability for the class label given the image and support set.

### 3.2 Loss Function and Training

Let $\mathcal{D}$ be a training set of image and label pairs. Our objective to minimize is:

$$\arg\min_\theta \sum_{(x,y)\in\mathcal{D}} \mathbb{E}_{\mathcal{S}\sim\mathcal{D}}\ L(f_\theta(x, \mathcal{S}), y), \tag{6}$$

where $L$ is the cross-entropy loss. This indirectly encourages similar-looking images to have higher similarity score, since more similar images will tend to have the same labels.

We optimize this objective using stochastic gradient descent (SGD) by sampling a mini-batch size $N_b$ of query-support pairs and computing gradients with respect to the mini-batch. Note that our model assumes no dependency between support set samples; indeed, we are free to sample any arbitrary $\mathcal{S}$ and query $x$, as long as the class of $x$ is within the support classes. Therefore, the support set size $N_s = |\mathcal{S}|$ is a hyperparameter analogous and orthogonal to the batch size $N_b$.

### 3.3 Inference

The ability to select and manipulate the support set at inference time unlocks a degree of flexibility not possible with parametric classification models. In particular, the label set $\mathcal{C}$ of the support set directly determines the label set of the resulting prediction. For example, if the user has prior knowledge of which classes a particular query is most likely to be, the support set can be restricted to only contain those classes.

Note that we assume the model has been trained with random support sets drawn from the training data for each mini-batch. To characterize the effect of the support set on inference, in our experiments we implement the following "inference modes":

1. **Full.** Use the entire training set: $\mathcal{S} = \mathcal{D}$.

2. **Random.** Sample uniformly at random over the dataset, such that each class is represented $k$ times: $\mathcal{S} \sim \mathcal{D}$ and $|\mathcal{S}| = k|\mathcal{C}|$.

3. **Cluster.** Given the trained model, we first compute the features for all the training datapoints. Then, we perform $k$-means clustering on the features of the training datapoints for each class. These $k$ cluster centroids are then used as the support features for each class. Note that in cluster mode, the support set does not correspond to observed datapoints.

4. **Closest cluster (CC).** Same as Cluster, except the support features correspond to real training datapoints that are closest to the cluster centroids in terms of Euclidean distance.

Each mode except Full assumes that each class is represented $k$ times in $\mathcal{S}$. One can, obviously, modify this assumption to reflect any class imbalance that might exist in the training data.

### 3.4 Support Influence

Koh & Liang (2017) define influence as the change in the loss as a result of upweighting a particular training point. With an NW head, quantifying the influence of a support image to the given query is straightforward and can be computed in closed-form. Indeed, the proposed support influence computes the change in the loss directly on the prediction vector itself (and not indirectly through the model parameters $\theta$), and furthermore does not require approximations, nor assumptions on convexity or differentiability.

Denote by $\mathcal{S}_{-z_s} = \mathcal{S} \setminus z_s$, i.e. the support set with an element removed. For a fixed query $x$:

$$f(x, \mathcal{S}_{-z_s}) = \frac{f(x, \mathcal{S}) - w(x, x_s)\vec{y}_s}{1 - w(x, x_s)}. \tag{7}$$

We define the support influence of $z_s$ on $z$ as the change in the cross-entropy loss incurred from removing a support element $z_s$:

$$\mathcal{I}(z, z_s) = L(f(x, \mathcal{S}_{-z_s}), y) - L(f(x, \mathcal{S}), y) = \log\left(\frac{f^y - f^y w(x, x_s)}{f^y - w(x, x_s)\mathbb{1}_{\{y=y_s\}}}\right), \tag{8}$$

where superscript denotes indexing into a vector and $\mathbb{1}$ is the indicator function. The derivation is provided in the Appendix.

## 4 Experiments

### 4.1 Experimental Setup

**Datasets.** We experiment with an array of computer vision datasets with different class diversity and training set size. For general image classification, we experiment with Cifar-100 (Krizhevsky, 2009). For fine-grained image classification with small to medium training set size (less than 12k training samples), we experiment with CUB-200-2011 (Bird-200) (Wah et al., 2011), Stanford Dogs (Dog-120) (Khosla et al., 2011), Oxford Flowers (Flower-102) (Nilsback & Zisserman, 2008), and FGVC-Aircraft (Aircraft-100) (Maji et al., 2013). For a large-scale fine-grained image classification task (500k training samples), we experiment with iNaturalist-10k (Grant Van Horn, 2021). Additional dataset details are provided in Appendix A.1.

**Network Architecture and Hyperparameters.** For the feature extractor $g_\theta$, we use a convolutional neural network (CNN) followed by a linear projection to an embedding space of dimension $d$. For Cifar-100, Bird-102, Dog-120, Flowers-102, and Aircraft-100 dataset, we experiment with the ResNet-18 (He et al., 2015) and DenseNet-121 (Huang et al., 2016) architectures. For iNaturalist-10k, we experiment with the ResNet-50 architecture, following prior work (Zhou et al., 2019). For all feature extractors, we set $d = 128$ and $\tau = 1$. We train each model 3 times with different random initializations. Additional model and hyperparameter details are provided in Appendix A.2.

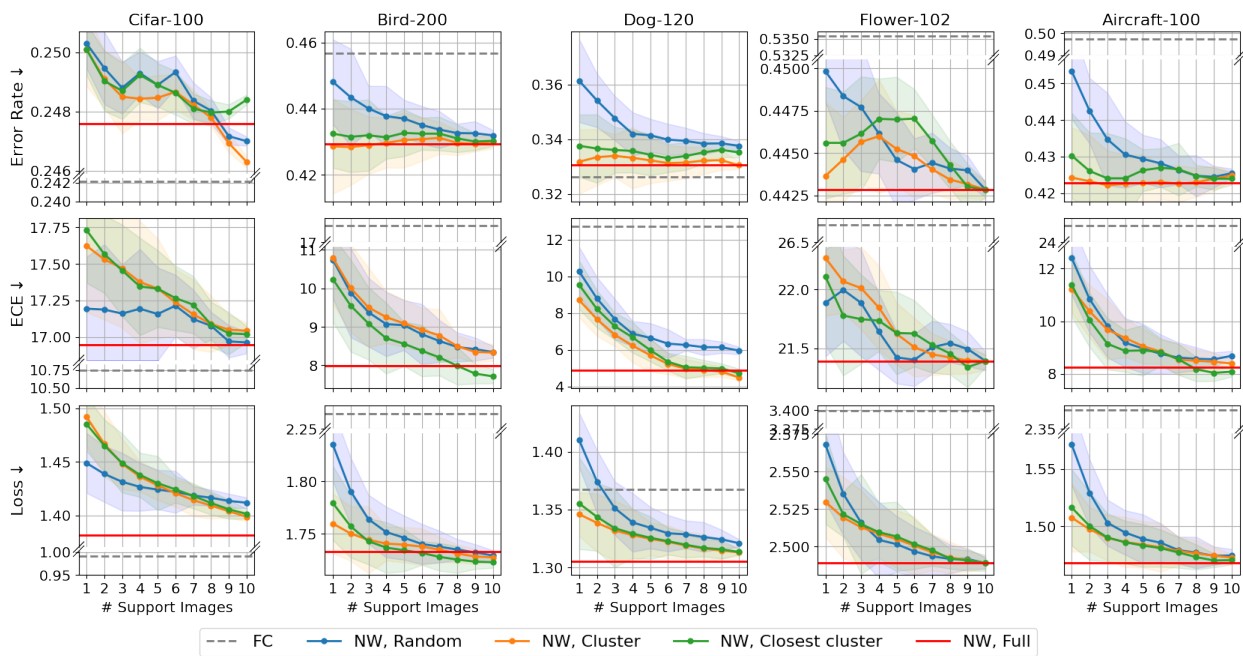

Figure 2: Model performance across different support set sizes for DenseNet-121. Full mode and parametric baseline are constant across support set size and are depicted as horizontal lines.

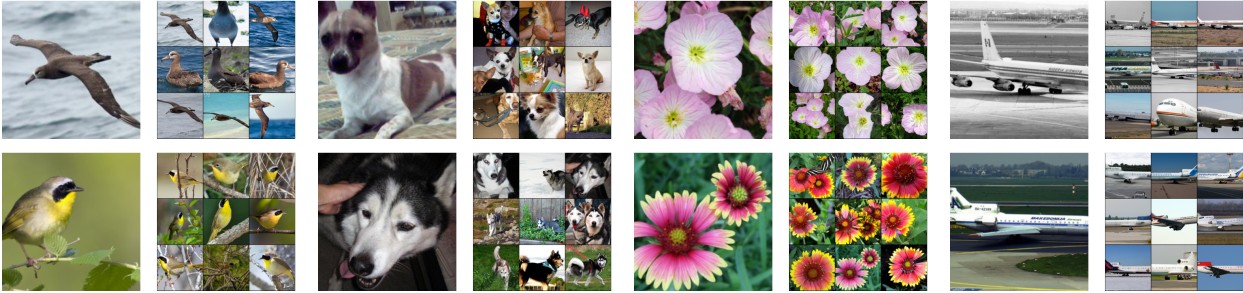

Figure 3: Corresponding images of embeddings closest to cluster centroids, for $k = 1$ and $k = 9$.

**Baseline.** We compare the NW head to a standard parametric model with a single-layer fully-connected (FC) head, and refer to it throughout this work as FC. For FC, we use the same feature extractor $g_\theta$ and keep all relevant hyperparameters identical to NW.

**Metrics.** We measure performance using error rate (the fraction of incorrectly classified test cases) and expected calibration error (ECE) (Guo et al., 2017; Naeini et al., 2015). ECE approximates the difference in expectation between confidence and accuracy, computed by partitioning the model predictions into bins and taking a weighted average of each bins' difference between confidence and accuracy.

## 4.2 Model Performance

Tables 1, 2, and 5 show error rate and ECE performance for all models and datasets, and Fig. 4 further shows reliability diagrams. We generally observe that the NW head achieves an error rate that is comparable and at times better than its parametric counterpart. Calibration is particularly improved in small to medium-sized datasets with relatively fine-grained differences between classes.

Table 1: Error rate ↓ (%). **Bold** is best and underline is second best. $k = 1$ for NW models.

|  | Model | Cifar-100 | Bird-200 | Dog-120 | Flower-102 | Aircraft-100 |
|---|---|---|---|---|---|---|
| ResNet-18 | FC | **26.86**$_{\pm 0.75}$ | 49.77$_{\pm 1.20}$ | **37.42**$_{\pm 1.15}$ | 47.00$_{\pm 0.27}$ | 49.41$_{\pm 0.88}$ |
|  | NW, Random | 28.41$_{\pm 1.00}$ | 51.45$_{\pm 0.97}$ | 45.33$_{\pm 1.05}$ | 43.86$_{\pm 0.22}$ | 41.76$_{\pm 1.03}$ |
|  | NW, Cluster | 27.66$_{\pm 1.15}$ | **46.60**$_{\pm 0.98}$ | 38.10$_{\pm 0.86}$ | **42.87**$_{\pm 0.32}$ | 39.94$_{\pm 1.15}$ |
|  | NW, CC | 27.82$_{\pm 0.90}$ | 47.76$_{\pm 1.07}$ | 39.00$_{\pm 0.99}$ | 43.70$_{\pm 0.33}$ | 39.99$_{\pm 1.09}$ |
|  | NW, Full | 27.58$_{\pm 0.87}$ | 46.98$_{\pm 0.71}$ | 38.07$_{\pm 0.91}$ | 42.97$_{\pm 0.30}$ | **39.93**$_{\pm 0.80}$ |
| DenseNet-121 | FC | **24.21**$_{\pm 0.96}$ | 45.68$_{\pm 0.87}$ | **32.62**$_{\pm 0.93}$ | 53.54$_{\pm 0.32}$ | 49.74$_{\pm 0.73}$ |
|  | NW, Random | 25.03$_{\pm 1.14}$ | 44.59$_{\pm 1.25}$ | 36.14$_{\pm 0.83}$ | 44.98$_{\pm 0.37}$ | 45.33$_{\pm 1.06}$ |
|  | NW, Cluster | 24.63$_{\pm 0.89}$ | **43.23**$_{\pm 0.72}$ | 33.42$_{\pm 1.03}$ | 44.36$_{\pm 0.34}$ | 42.42$_{\pm 1.09}$ |
|  | NW, CC | 24.84$_{\pm 0.97}$ | 43.68$_{\pm 1.05}$ | 33.96$_{\pm 0.89}$ | 44.56$_{\pm 0.28}$ | 43.02$_{\pm 0.86}$ |
|  | NW, Full | 24.76$_{\pm 0.89}$ | 43.33$_{\pm 1.33}$ | 33.07$_{\pm 1.18}$ | **44.28**$_{\pm 0.30}$ | **42.27**$_{\pm 0.84}$ |

Table 2: ECE ↓ (%). **Bold** is best and underline is second best. $k = 1$ for NW models.

|  | Model | Cifar-100 | Bird-200 | Dog-120 | Flower-102 | Aircraft-100 |
|---|---|---|---|---|---|---|
| ResNet-18 | FC | **15.40**$_{\pm 1.21}$ | 20.23$_{\pm 1.10}$ | 16.02$_{\pm 1.07}$ | 22.26$_{\pm 0.37}$ | 24.55$_{\pm 0.96}$ |
|  | NW, Random | 18.84$_{\pm 1.13}$ | 7.762$_{\pm 1.10}$ | 5.583$_{\pm 0.89}$ | 19.59$_{\pm 0.43}$ | 12.16$_{\pm 1.10}$ |
|  | NW, Cluster | 19.40$_{\pm 0.79}$ | 6.051$_{\pm 0.77}$ | 4.025$_{\pm 0.93}$ | 19.83$_{\pm 0.37}$ | 12.12$_{\pm 1.29}$ |
|  | NW, CC | 19.47$_{\pm 0.84}$ | 6.224$_{\pm 0.87}$ | 4.757$_{\pm 0.98}$ | 20.09$_{\pm 0.43}$ | 11.66$_{\pm 0.73}$ |
|  | NW, Full | 17.68$_{\pm 0.85}$ | **3.327**$_{\pm 0.65}$ | **4.256**$_{\pm 1.17}$ | **19.10**$_{\pm 0.41}$ | **9.235**$_{\pm 1.05}$ |
| DenseNet-121 | FC | **10.74**$_{\pm 0.77}$ | 17.67$_{\pm 1.11}$ | 12.74$_{\pm 0.92}$ | 26.79$_{\pm 0.41}$ | 25.03$_{\pm 0.89}$ |
|  | NW, Random | 17.19$_{\pm 1.08}$ | 10.74$_{\pm 1.37}$ | 10.26$_{\pm 0.95}$ | 21.89$_{\pm 0.45}$ | 12.40$_{\pm 1.05}$ |
|  | NW, Cluster | 17.62$_{\pm 0.96}$ | 10.78$_{\pm 0.97}$ | 8.709$_{\pm 0.77}$ | 22.27$_{\pm 0.47}$ | 11.23$_{\pm 1.14}$ |
|  | NW, CC | 17.73$_{\pm 1.03}$ | 10.23$_{\pm 1.19}$ | 9.535$_{\pm 0.99}$ | 22.11$_{\pm 0.37}$ | 11.39$_{\pm 0.86}$ |
|  | NW, Full | 16.95$_{\pm 0.95}$ | **7.997**$_{\pm 0.68}$ | **4.892**$_{\pm 0.75}$ | **21.39**$_{\pm 0.38}$ | **8.248**$_{\pm 1.11}$ |

Table 4 shows inference times on Bird-200 for both parametric and the NW-head models, where we assume that all support embeddings, cluster centroids, and closest cluster embeddings have been precomputed. We observe that inference times are nearly identical to the FC head, except Full. On a small dataset like Bird-200, Full corresponds to a two-fold increase in computation time.

In Fig. 2, we plot various metrics across support set size $k$. Generally, performance improves as support set size increases. We attribute this to the fact that more support images leads to more robust predictions, as more support examples per class can average out potential errors. At the limit, Full mode typically performs the best, at the cost of inference time.

Cluster mode often improves over Random mode and is on par with Full mode (at larger $k$) while having favorable inference time. However, this comes at the cost of loss of interpretability, since the cluster centroids do not correspond to actual images in the support set. Closest cluster addresses this deficiency, often with minimal sacrifice to performance compared to Cluster.

Fig. 3 visualizes the support images used in Closest cluster for two classes, with $k = 1$ and $k = 9$. These images are the prototypical examples of the class and can be interpreted as a summary or compression of the class and dataset. We observe that for $k = 1$, the support image is a prototypical example of the class. For $k = 9$, the images have a variety of poses and backgrounds, allowing the model to have a higher chance of computing similarities with a wide range of class prototypes. Further research might explore leveraging these prototypes.

Overall, our experiments indicate that the NW head has advantages in accuracy and calibration in the datasets we tested. The NW head pulls ahead in performance particularly for fine-grained classification tasks, and pulls ahead even further for small-medium sized datasets. Computational load is higher than FC,

Table 3: Error rate ↓ / ECE ↓ (%) for label smoothing (LS) and temperature scaling (TS) for ResNet-18. **Bold** is best. Cluster and $k = 1$ for NW models.

| Model | Bird-200 | Dog-120 | Flower-102 | Aircraft-100 |
|---|---|---|---|---|
| FC-LS | $45.36_{\pm 1.03}$ / $15.83_{\pm 0.98}$ | $\mathbf{37.58}_{\pm 1.14}$ / $10.79_{\pm 0.83}$ | $46.35_{\pm 0.45}$ / $3.84_{\pm 0.38}$ | $48.72_{\pm 0.91}$ / $7.16_{\pm 0.97}$ |
| NW-LS | $\mathbf{45.10}_{\pm 0.82}$ / $\mathbf{2.097}_{\pm 0.86}$ | $38.29_{\pm 0.63}$ / $\mathbf{2.24}_{\pm 1.08}$ | $\mathbf{43.23}_{\pm 0.40}$ / $\mathbf{1.99}_{\pm 0.29}$ | $\mathbf{45.76}_{\pm 1.01}$ / $\mathbf{3.61}_{\pm 0.85}$ |
| FC-TS | $49.77_{\pm 1.20}$ / $2.711_{\pm 0.92}$ | $\mathbf{37.42}_{\pm 1.15}$ / $2.208_{\pm 0.97}$ | $47.00_{\pm 0.27}$ / $4.426_{\pm 0.31}$ | $49.41_{\pm 0.88}$ / $\mathbf{4.423}_{\pm 1.00}$ |
| NW-TS | $\mathbf{46.60}_{\pm 0.98}$ / $\mathbf{2.460}_{\pm 0.89}$ | $38.10_{\pm 0.86}$ / $\mathbf{1.911}_{\pm 0.77}$ | $\mathbf{42.87}_{\pm 0.32}$ / $2.693_{\pm 0.40}$ | $\mathbf{39.94}_{\pm 1.15}$ / $5.191_{\pm 1.10}$ |

| Model | Time $(10^{-3}$ sec) |
|---|---|
| FC | $4.00_{\pm 0.05}$ |
| NW, $k = 1$ | $4.05_{\pm 0.08}$ |
| NW, $k = 10$ | $4.78_{\pm 0.09}$ |
| NW, Full | $7.76_{\pm 0.08}$ |

Table 4: GPU inference times for Bird-200 and ResNet-18.

| Model | Error Rate | ECE |
|---|---|---|
| FC | $43.7_{\pm 0.98}$ | $15.0_{\pm 0.58}$ |
| NW, k=1 | $43.1_{\pm 1.70}$ | $16.7_{\pm 0.73}$ |
| NW, k=10 | $42.5_{\pm 1.42}$ | $13.4_{\pm 0.83}$ |

Table 5: iNaturalist-10k results on ResNet-50. Cluster mode for NW.

but is only two times higher for our most expensive inference mode on smaller datasets with less than 10k training samples.

### 4.2.1 Improving Calibration

In this section, we explore whether existing calibration techniques originally proposed for parametric models can further improve calibration for NW models. Note that improved calibration is not specifically enforced in our architecture or training procedure; thus, many effective calibration techniques proposed in the literature can be readily applied to the NW head.

We explore two popular and effective techniques: label smoothing (LS) and temperature scaling (TS). Label smoothing (Szegedy et al., 2015) replaces one-hot labels during training with soft labels which are a weighted average of the one-hot labels and the uniform distribution. Temperature scaling (Guo et al., 2017) is a post-training technique which optimizes the temperature of the softmax with respect to the loss on the validation set (and freezing all trained parameters). For the NW head, the temperature is analogous to $\tau$ in Eq. 5. Additional details are provided in A.3.

We show the results in Table 3. In general, we find that the NW head outperforms the FC head under both LS and TS techniques. In particular, the simple, post-training technique of TS is a simple strategy that further improves our calibration performance.

### 4.3 Interpretability and Explainability

In the following section, we restrict our attention to Full mode and explore the interpretability and explainability aspects of the NW head.

### 4.3.1 Interpreting the Weights

Visualizing the support images ranked by weight $w$ can give insight to users about how the model is making a prediction. For cases where the model is unsure, visualizing the support images along with confidence scores can uncover where the model is unsure and potentially allow for user intervention.

In Fig. 6, we plot three examples of a query image and the top support set images ranked by weight $w$, along with the top 3 softmax predictions. We notice significant visual similarity between the query image and the top support set images. Notably, the second query of a "cardinal" is incorrectly classified by the model. We

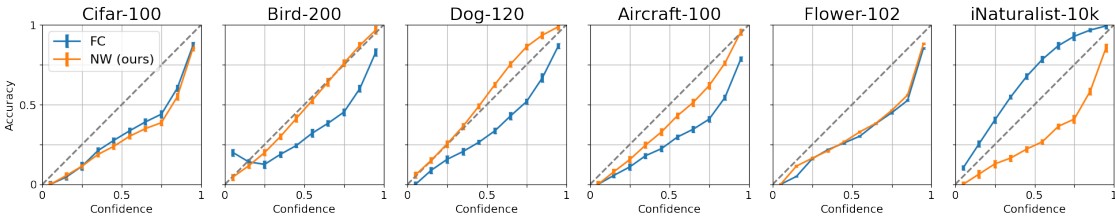

Figure 4: Reliability diagrams. NW generally has an advantage in calibration performance on small to medium-sized datasets.

notice that the model is confused between the top 2 classes in the support set. The model thinks the third query of "barbeton daisy" is most similar to "sunflowers", possibly due to the similarity in color and shape. A user can look at the support set and potentially intervene, either by correcting the model's prediction or removing confounding elements from the support set. An automated method for flagging conspicuous predictions could be based on confidence scores. Additional visualizations are provided in Appendix A.7.

**Comparing against FC features.** In Appendix A.8, we compare against ranking images by Euclidean distance in the feature space for the baseline FC model, and show that the degree of semantic similarity between the query and top most similar images is lower than the NW head. In Fig. 5, we quantify this by computing the percentage of top support set labels which match the query label. The NW model has a higher degree of semantic similarity with its top support images. We attribute this to the fact that semantic similarity is explicitly enforced in the NW head (i.e. features from the same class are encouraged to be close), while this is not enforced in the FC head.

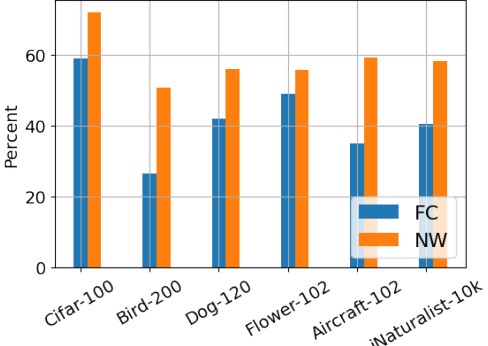

Figure 5: Percentage of top support set labels which match the query label. Top labels are found by ranking support images by negative Euclidean distance and taking the top 10 images. The NW model has a higher degree of semantic similarity with its top support images.

### 4.3.2 Explaining Model Behavior with Support Influence

Support influence can be used to debug and diagnose issues in otherwise black-box neural networks. In this section, we explore the use of support influence in helping to determine helpful and harmful training examples. Note that the support influence requires knowledge of the ground truth label for the query image and thus cannot be used during inference/test time.

By ranking and visualizing the support set by support influence, we can see the support images which are most helpful (highest $\mathcal{I}(z, z_s)$ values) and most harmful (lowest $\mathcal{I}(z, z_s)$ values) in predicting the given query $z$. In Fig. 7, we visualize three query images and their top 5 most helpful and top 5 most harmful support images. The top images are "helpful" in the sense that they belong to the same class as the query and look visually similar. The bottom images are "harmful" since, despite the fact that they are visually similar, they are a different class. For the first dog example, we see that "redbone" and "walker hound" breeds are harmful examples for the "beagle" query image.

The bird and flower queries are the same query images from Fig. 6, this time ranked by influence. For the bird, we see that helpful and harmful support images separates the two classes that had the highest weight and were confusing the model in Fig. 6. For the flower, we notice that the most helpful support images are from the correct class but have different colors, whereas the images which the model thinks are most similar to the query are in fact the most harmful. From this analysis, it is evident that the model is using the shape and color to reason about the query image, which are confounding features for this query.

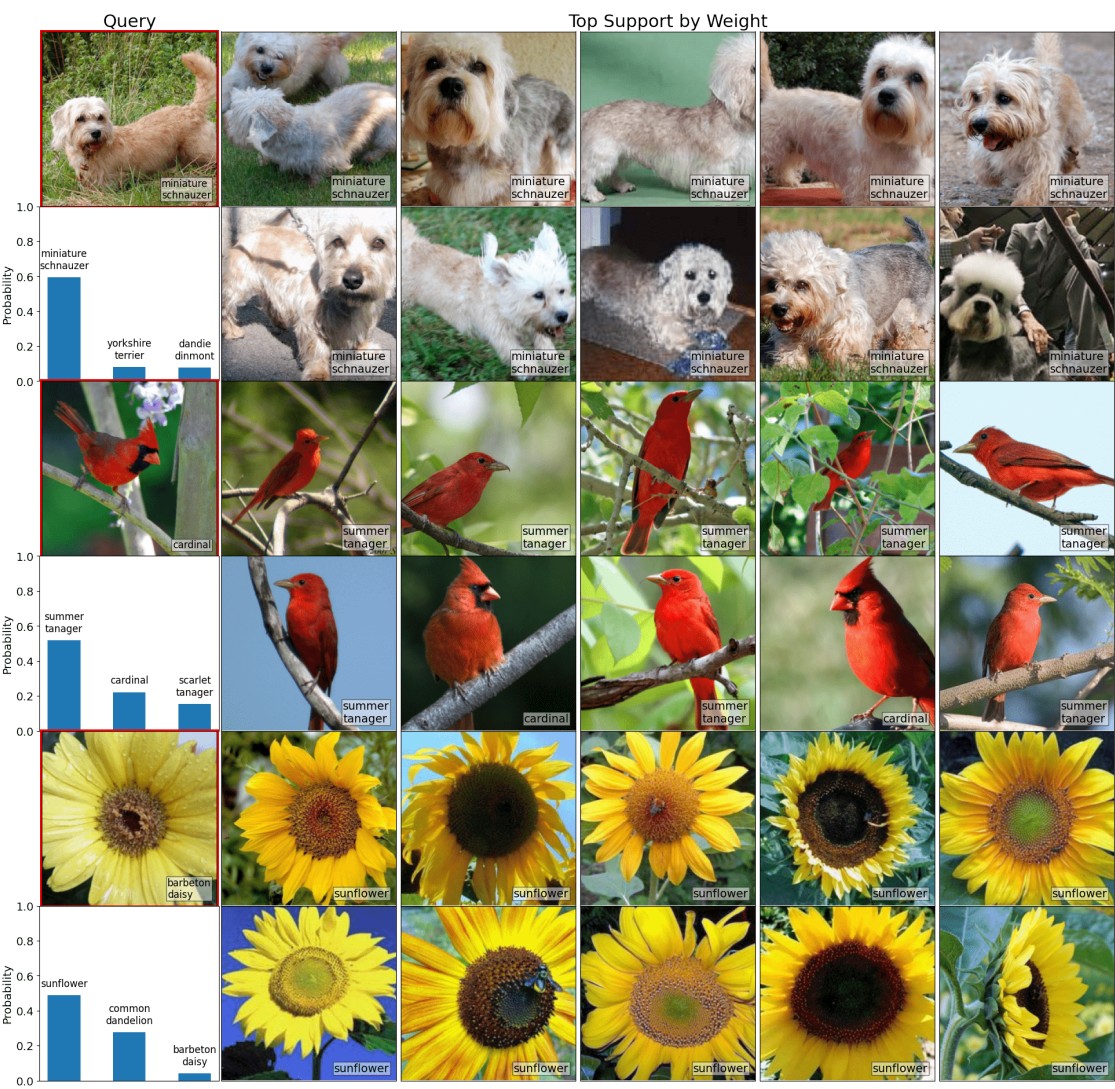

Figure 6: Examples of correctly (dog) and incorrectly (bird and flower) classified query images and their top support images ranked by $w$. Ordering is left to right, top to bottom.

## 5    Discussion

In this study, we find that the NW head has comparable to improved accuracy on all datasets we tested on. For calibration, NW head had the most improvement on data-starved scenarios with relatively fine-grained classification tasks. Indeed, on the small to medium-sized datasets (Bird-200, Dog-102, Flower-102, and Aircraft-100), we find the most dramatic improvement in calibration on all model variants tested. This calibration improvement is less pronounced in a large dataset like iNaturalist-10k, and the improvement disappears (although is still comparable) for more diverse datasets like Cifar-100[1]. Intuitively, this could be attributed to the fact that classification tasks with lower inter-class variability will tend to have higher similarities across classes, thus leading to better-calibrated uncertainties in the model predictions. On the other hand, tasks with high inter-class variability might be more suited for learning separating hyperplanes, as is the case with FC. We believe that an NW model is well-suited for applications where data is not

---

[1]Recent work has shown that large datasets are not as prone to poor calibration (Minderer et al., 2021)

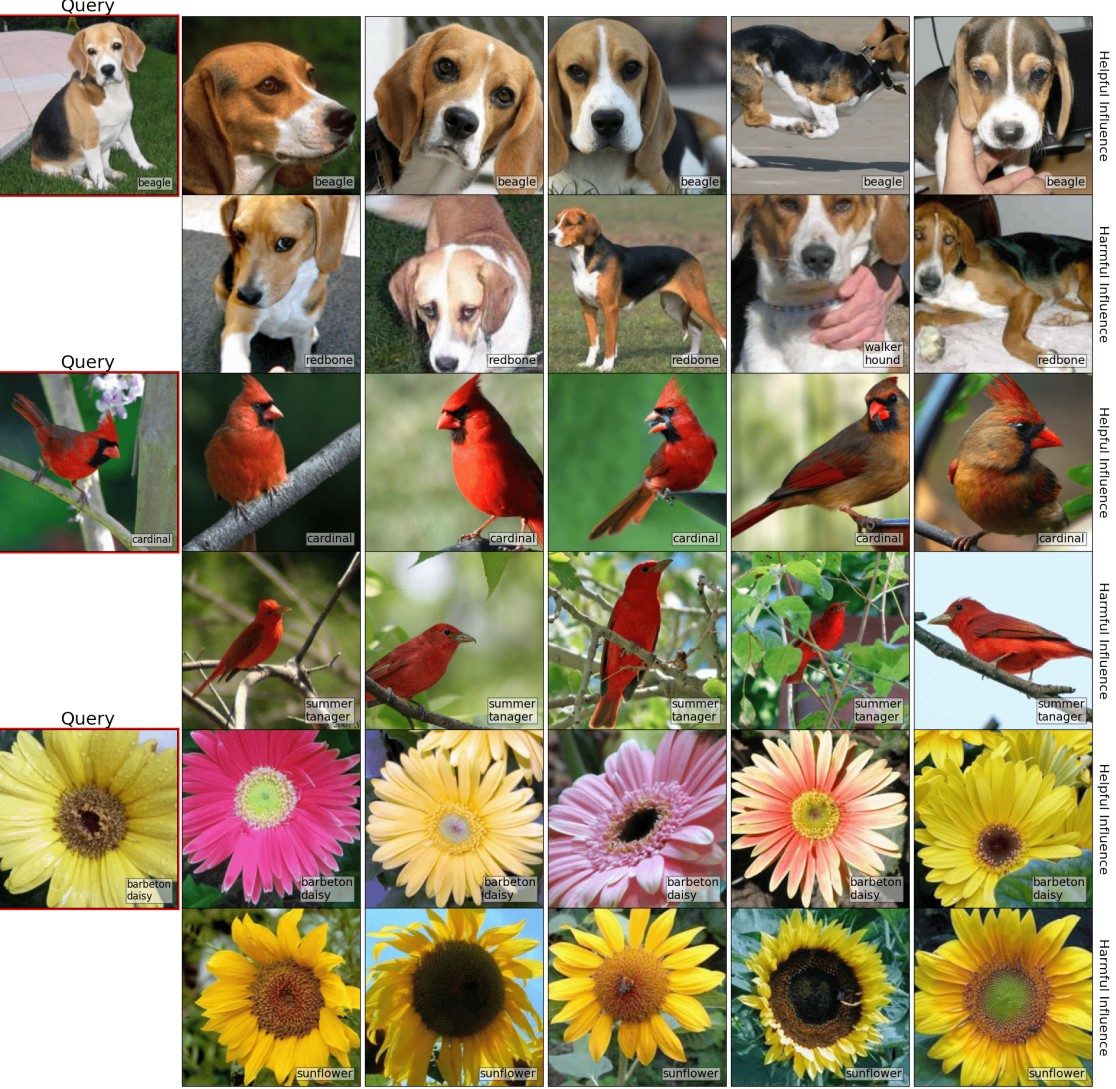

Figure 7: Query images and support images separated by most helpful influence (top row) and most harmful influence (bottom row).

abundant and differences between classes are subtle, like in medical imaging. Due to its simplicity, future work may explore combining the FC and NW heads in some fashion to further improve model performance.

The computational load of the NW head for Full mode is higher than the FC head, although this load is less for smaller datasets. However, flexibility at inference-time and interpretability are advantages of the NW head. In this study, we explore several choices of inference mode which each have advantages and trade-offs. In particular, we find that Cluster mode can be a sufficient replacement to Full mode (Fig. 2), while being computationally cheaper (Table 4). Nevertheless, Full mode allows user practitioners to interpret and debug a model's behavior (Section 4.3). Thus, one can imagine a workflow for the practitioner which would involve using Cluster mode to perform efficient inference, and then turning to Full mode on a select few (potentially problematic) test queries to understand model behavior. We stress that this flexibility at inference time is not possible with FC baseline models, which have the clear advantage in terms of computational efficiency.

## 6 Conclusion

We presented the Nadaraya-Watson (NW) head as a general-purpose, flexible, and interpretable building block for deep learning. The NW head predicts the label for a given query by taking the weighted sum of labels in a support set. We show that the NW head can be an efficient replacement for the fully-connected layer and offer good accuracy with excellent calibration, particularly in data-starved and fine-grained classification tasks. Existing calibration techniques proposed for parametric models can lead to further calibration gains. Additionally, the NW head is interpretable in the sense that the weights correspond to the contribution of each support image on the query image. Finally, we define support influence, and show that this can be used as a tool to explain model behavior by highlighting helpful and harmful support images.

## 7 Acknowledgements

This work was supported by NIH grants R01LM012719, R01AG053949, and RF1 MH123232; the NSF NeuroNex grant 1707312, and the NSF CAREER 1748377 grant.

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

Table 6: Summary of Datasets

| Dataset | Description | # Train | # Test | # Classes |
|---|---|---|---|---|
| Cifar-100 (Krizhevsky, 2009) | Image classification | 50,000 | 10,000 | 100 |
| Bird-200 (Wah et al., 2011) | Bird species classification | 5,794 | 5,994 | 200 |
| Dog-120 (Khosla et al., 2011) | Dog species classification | 12,000 | 8,580 | 120 |
| Flower-102 (Nilsback & Zisserman, 2008) | Flower species classification | 2,040 | 6,149 | 102 |
| Aircraft-100 (Maji et al., 2013) | Aircraft classification | 3,334 | 3,333 | 100 |
| iNaturalist (Grant Van Horn, 2021) | Species classification | 500,000 | 100,000 | 10000 |

# A  Appendix

## A.1  Datasets

In this section, we give an overview of the datasets used in this study. Table 6 gives a summary of all datasets, including description, train/test sizes, and number of classes. Cifar-100 has input image sizes of 32x32, while all other datasets are resized to 256x256 and cropped to 224x224. For Cifar-100, Flower-102, Aircraft-100, and iNaturalist-10k, we use the implementation in the `torchvision` package. For iNaturalist-10k, we use the "mini" training dataset from the 2021 competition, which has balanced images per class, and test on the validation set (the test set is not provided). For Bird-200[2] and Dog-120[3], we pull the train/test splits from the dataset websites.

## A.2  Models and Training Details

For all proposed models and baselines, we use models implemented in the `torchvision` package, with randomly initialized weights. Following the setup of Yun et al. (2020), we use standard ResNet-18 with 64 filters and DenseNet-121 with a growth rate of 32 for image size $224 \times 224$. For Cifar-100, we use PreAct ResNet-18 (He et al., 2016), which modifies the first convolutional layer4 with kernel size $3 \times 3$, strides 1 and padding 1, instead of the kernel size $7 \times 7$, strides 2 and padding 3, for image size $32 \times 32$. We use DenseNet-BC structure (Huang et al., 2016), and the first convolution layer of the network is also modified in the same way as in PreAct ResNet-18 for image size $32 \times 32$. We use the standard data augmentation technique of random flipping and cropping for all experiments.

For NW training, we use SGD with momentum 0.9, weight decay 1e-4, and an initial learning rate of 1e-3. The learning rate is divided by 10 after 20K and 30K gradient steps, and training stops after 40K gradient steps. For all datasets except iNaturalist-10k, we set the randomly sampled support size to be $N_s = 10$ for all datasets, and the mini-batch size to be $N_b = 32$ Cifar-100 and $N_b = 4$ for Bird-200, Dog-120, Flower-102, and Aircraft-100. That is, we sample a unique support set for each query, following Eq. 6. For iNaturalist-10k, we use a mini-batch size $N_b = 128$ and support size $N_s = 250$. We sample a single support set for each mini-batch (instead of each query) for computational efficiency. Effectively, this makes the total number of images per mini-batch to be $N_s + N_b$ instead of $N_s N_b$. Note that there is very little effect of this change on training because the number of unique pairwise similarities is preserved.

For FC training, we train for 200 epochs, use an initial learning rate of 0.1, and divide the learning rate by 10 after epochs 100 and 150. We found that this setting yielded better results for the baseline models. All other hyperparameters are identical to the NW setup.

All training and inference is done on an Nvidia A6000 GPU and all code is written in Pytorch[4].

---

[2]http://www.vision.caltech.edu/datasets/cub_200_2011/
[3]http://vision.stanford.edu/aditya86/ImageNetDogs/
[4]Our code is available at https://github.com/alanqrwang/nwhead.

### A.3 Details on Calibration Experiments

To produce results in Section 4.2.1, we follow methods and hyperparameters from the original papers on temperature scaling (TS) (Guo et al., 2017) and label smoothing (LS) (Szegedy et al., 2015).

For LS, we modify all ground truth labels by taking the weighted average between the one-hot label and the uniform distribution:

$$\vec{y}_{ls} = (1 - \epsilon)\vec{y} + \frac{\epsilon}{C}\mathbf{1}, \tag{9}$$

where $C$ is the number of classes and $\mathbf{1}$ is a vector of all ones. We choose the weight used in the original paper, $\epsilon = 0.1$.

For TS, we sweep $\tau$ over 100 values linearly spaced between 0.5 and 3. We select the optimal $\tau$ which minimizes loss on the validation set.

### A.4 Derivation of Support Influence

We derive the expression in Eq. (8). For brevity, let $f = f(x, \mathcal{S})$ and $f_- = f(x, \mathcal{S}_{-z_s})$.

$$\mathcal{I}(z, z_s) = L(f_-, y) - L(f, y) = \sum_{c=1}^{C} \vec{y}^c \log f^c - \sum_{c=1}^{C} \vec{y}^c \log f_-^c$$

$$= \log f^y - \log f_-^y.$$

If $y = y_s$, then $w(x, x_s)$ must be subtracted from $f^y$ and re-normalized to a valid probability. Otherwise, $f^y$ is simply re-normalized:

$$f_-^y = \begin{cases} \frac{f^y - w(x, x_s)}{1 - w(x, x_s)}, & \text{if } y = y_s \\ \frac{f^y}{1 - w(x, x_s)}, & \text{else.} \end{cases}$$

Thus,

$$\mathcal{I}(z, z_s) = \log f^y - \log\left(\frac{f^y - w(x, x_s)\mathbb{1}_{\{y=y_s\}}}{1 - w(x, x_s)}\right)$$

$$= \log\left(\frac{f^y - f^y w(x, x_s)}{f^y - w(x, x_s)\mathbb{1}_{\{y=y_s\}}}\right).$$

### A.5 Connection between Support Influence and Weights

We prove that ranking by helpful/harmful examples via the support influence is equivalent to ranking by the weights for correct and incorrect classes. Let us start from the definition of support influence in Eq. (8):

$$\mathcal{I}(z, z_s) = L(f(x, \mathcal{S}_{-z_s}), y) - L(f(x, \mathcal{S}), y) = \log\left(\frac{f^y - f^y w(x, x_s)}{f^y - w(x, x_s)\mathbb{1}_{\{y=y_s\}}}\right).$$

where $0 \leq w \leq f^y \leq 1$. Note that $w \leq f^y$ because they are related by Eq. (3).

Separate into two cases:

$$\mathcal{I}(z, z_s) = \begin{cases} \log\left(\frac{1 - w(x, x_s)}{1 - w(x, x_s)/f^y}\right), & \text{if } y = y_s \\ \log(1 - w(x, x_s)), & \text{else.} \end{cases}$$

In the first case where $y = y_s$, $\mathcal{I}(z, z_s)$ increases as $w$ increases, with the constraints on $w$ and $f^y$.

In the second case where $y \neq y_s$, $\mathcal{I}(z, z_s)$ decreases as $w$ increases.

In words, the support influence increases as the weight increases for the correct class, and decreases as the weight increases for the incorrect class. Thus, the support influence can be interpreted as simply ranking by the weights for correct and incorrect classes.

Table 7: Error rate ↓ for various settings of $(N_b, N_s)$, assuming a fixed $N_b N_s$. Cluster and $k = 1$ for NW models.

| $(N_b, N_s)$ | (1, 40) | (2, 20) | (4, 10) | (5, 8) | (8, 5) | (10, 4) | (20, 2) |
|---|---|---|---|---|---|---|---|
| Error rate | 54.7 | 51.2 | 46.6 | 46.8 | 47.1 | 51.0 | 58.2 |

### A.6   Analysis of Batch Size and Support Set Size

In this section, we provide a hyperparameter sweep of $N_s$ and $N_b$ and present error rate results on the Bird-200 dataset and ResNet-18. As discussed in  A.2, the total number of images per mini-batch is defined by $N_s N_b$. Common practice in machine learning dictates that the total number of images should be maximized in order to obtain best convergence. Thus, in this analysis, we keep $N_s N_b$ to be fixed to 40 (same as our reported results), and sweep the values with this constraint.

We find that there is a sweet spot around (4,10) and (5,8). We hypothesize that low batch size $N_b$ hurts convergence, as is typical for SGD optimization. On the other hand, low $N_s$ may be non-ideal because the model does not have a sufficient number of "negative" samples to compare against. With that said, the performance is not particularly sensitive with respect to reasonable values of $N_s$. We believe that balancing this tradeoff is important to maximizing performance on a given dataset.

## A.7 More NW Visualizations

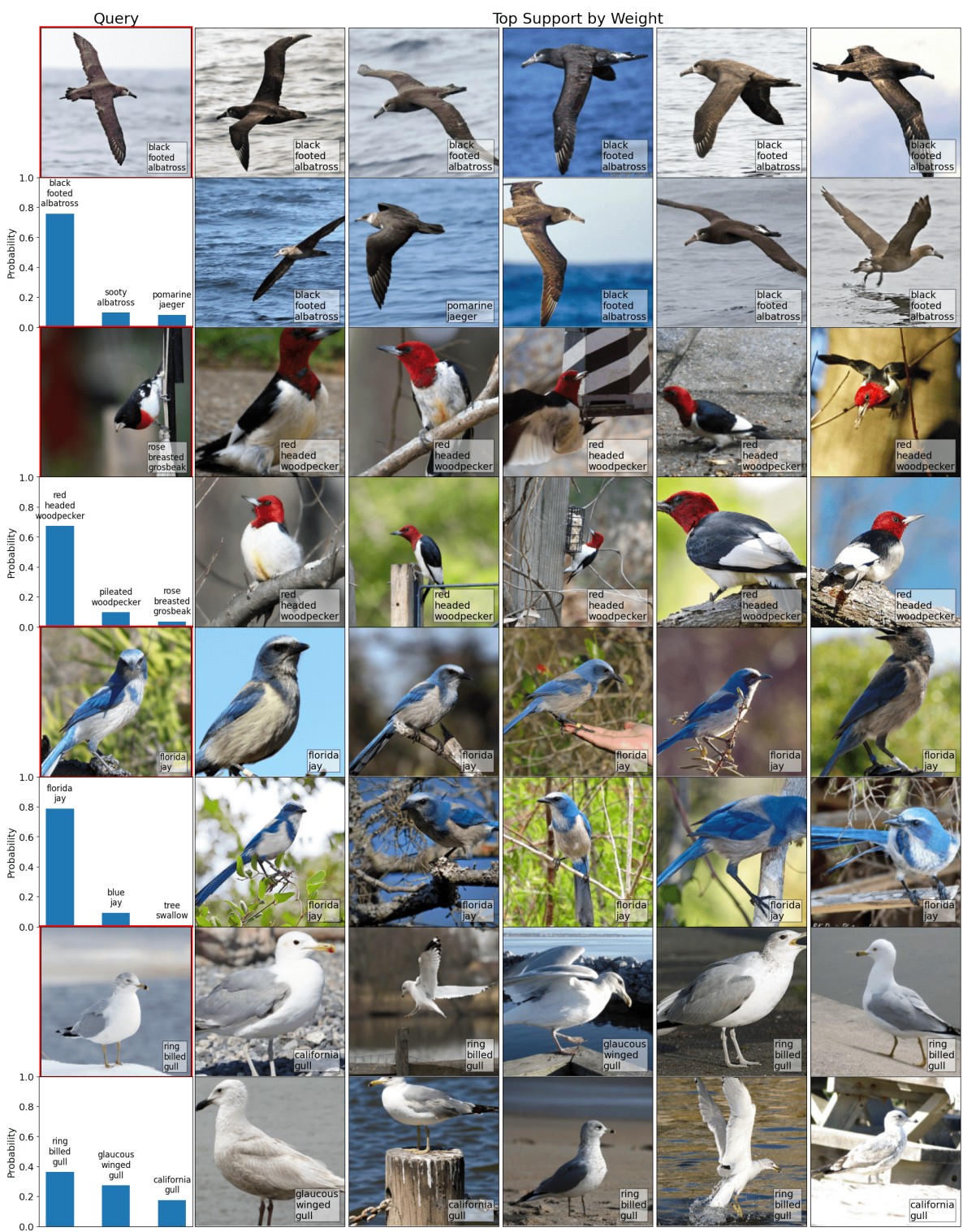

Figure 8: Query image, top 3 predicted probabilities, and top support ranked by weight for Bird dataset.

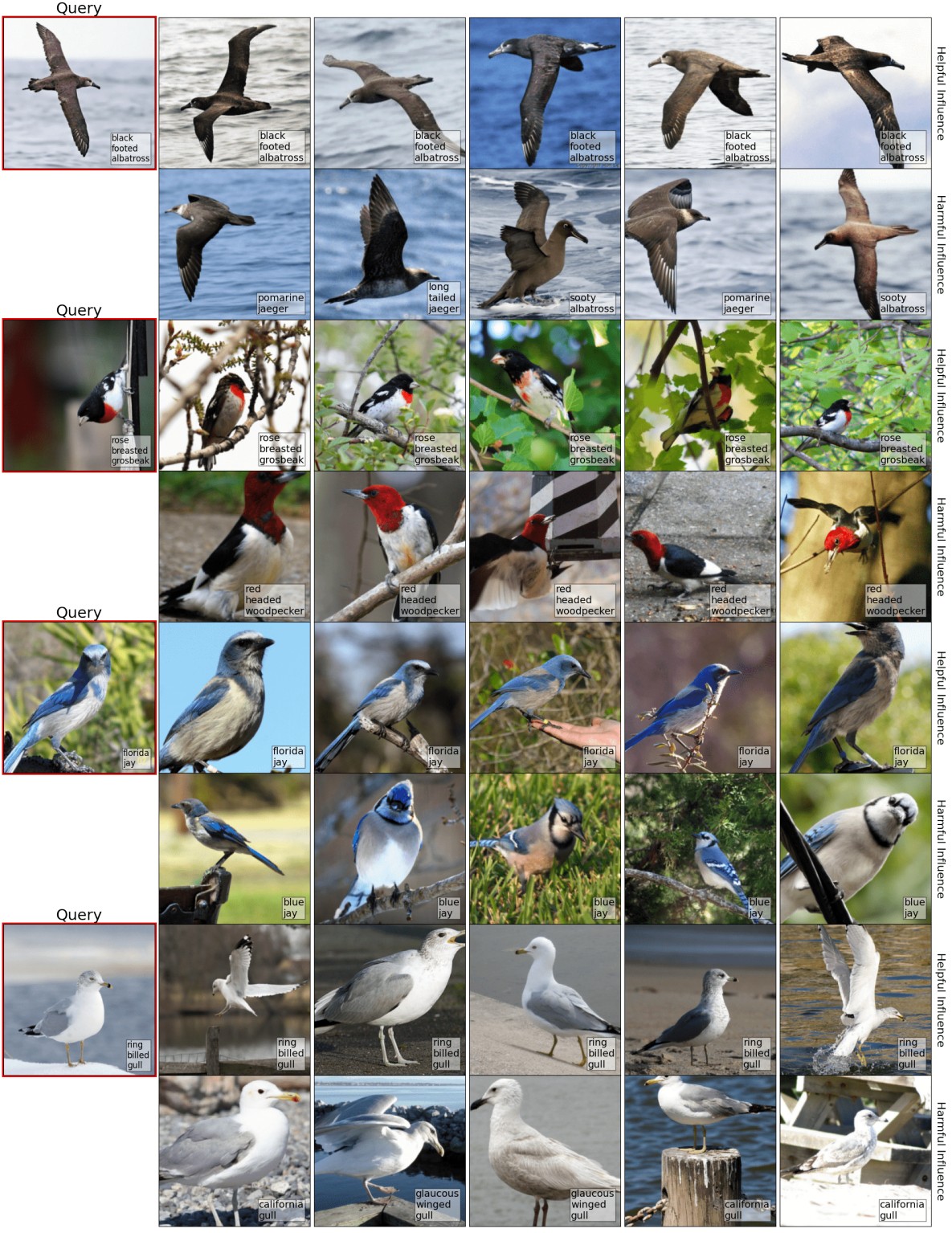

Figure 9: Query image, top helpful support images, and top harmful support images ranked via support influence for Bird dataset.

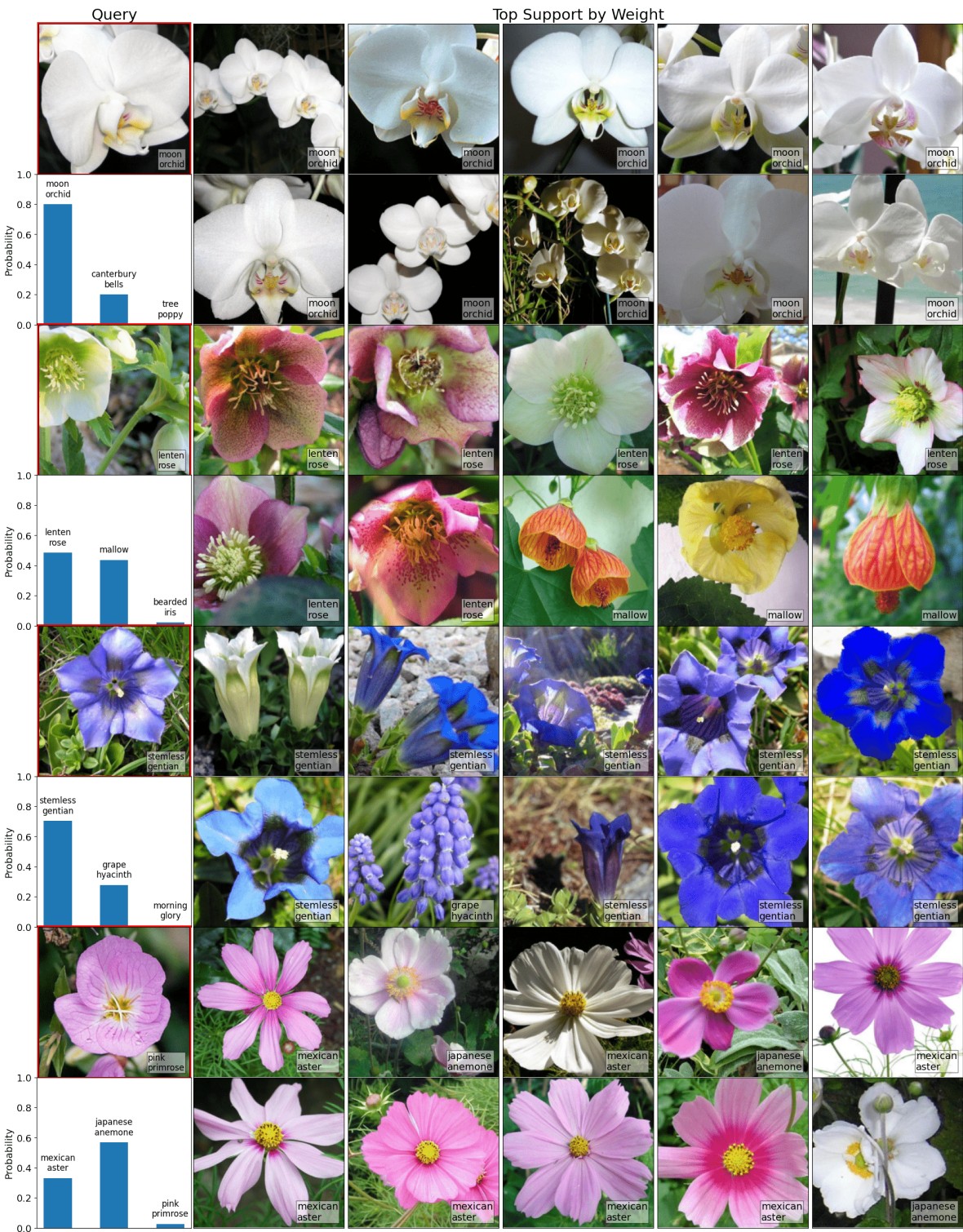

Figure 10: Query image, top 3 predicted probabilities, and top support ranked by weight for Flower dataset.

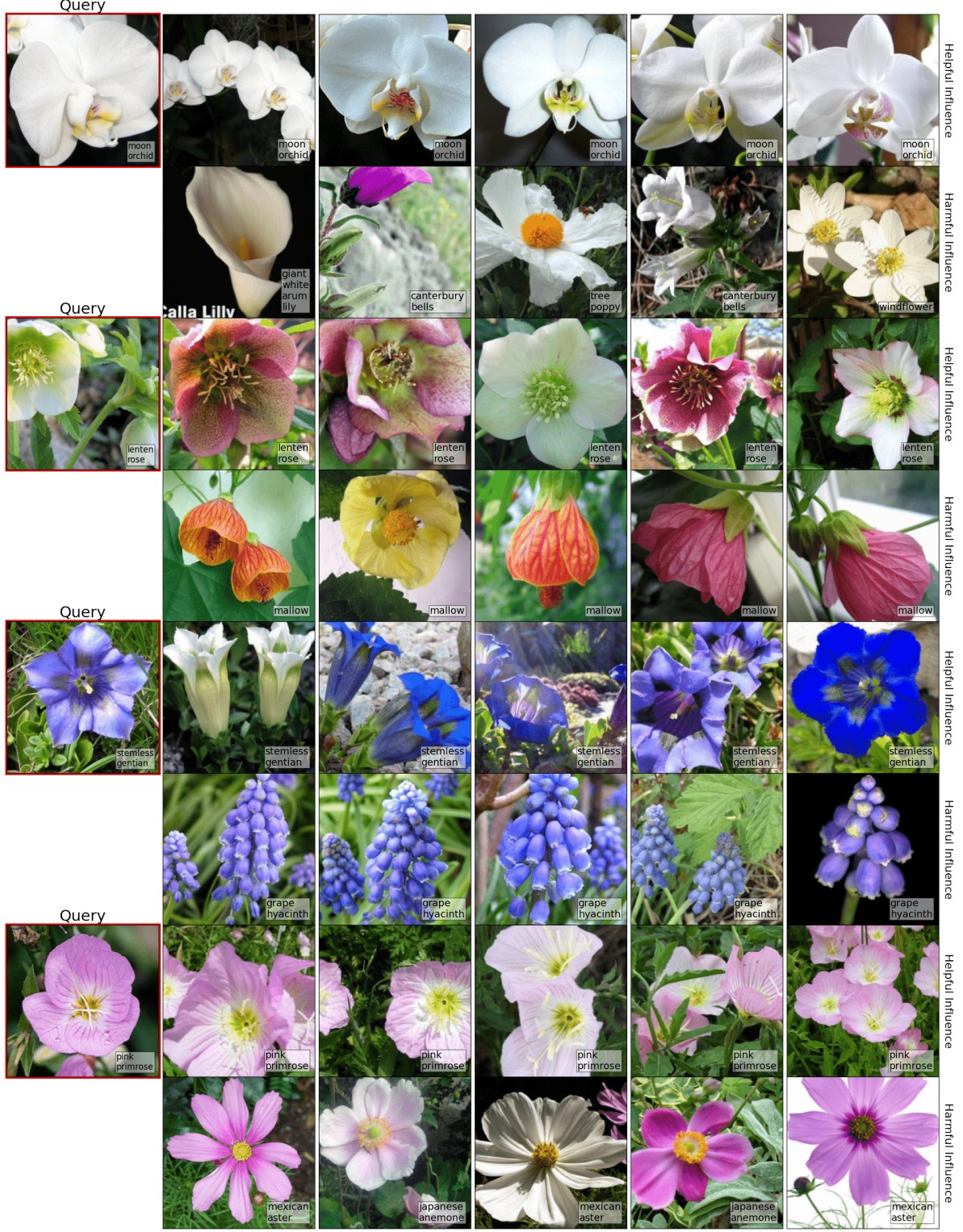

Figure 11: Query image, top helpful support images, and top harmful support images ranked via support influence for Flower dataset.

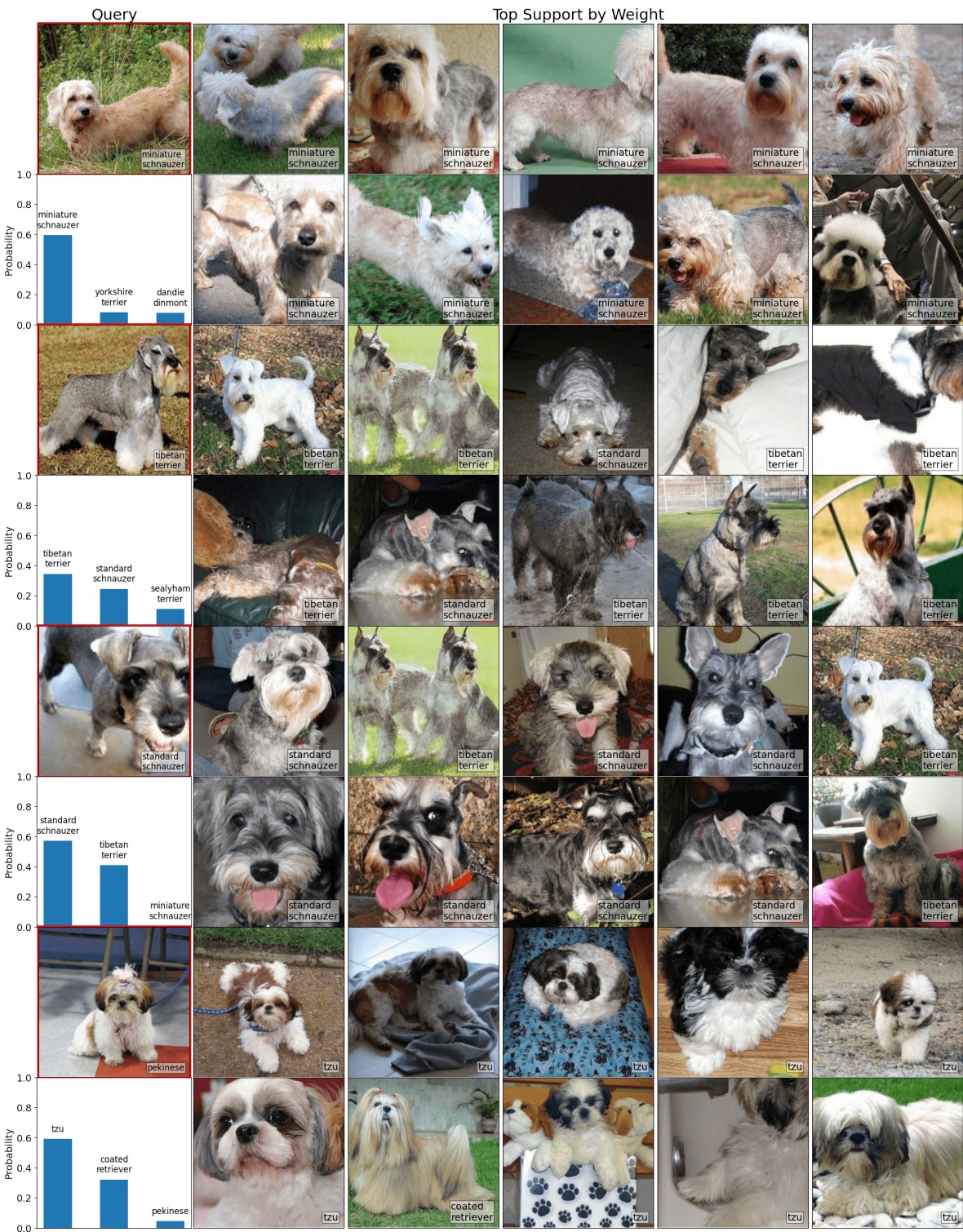

Figure 12: Query image, top 3 predicted probabilities, and top support ranked by weight for Dog dataset.

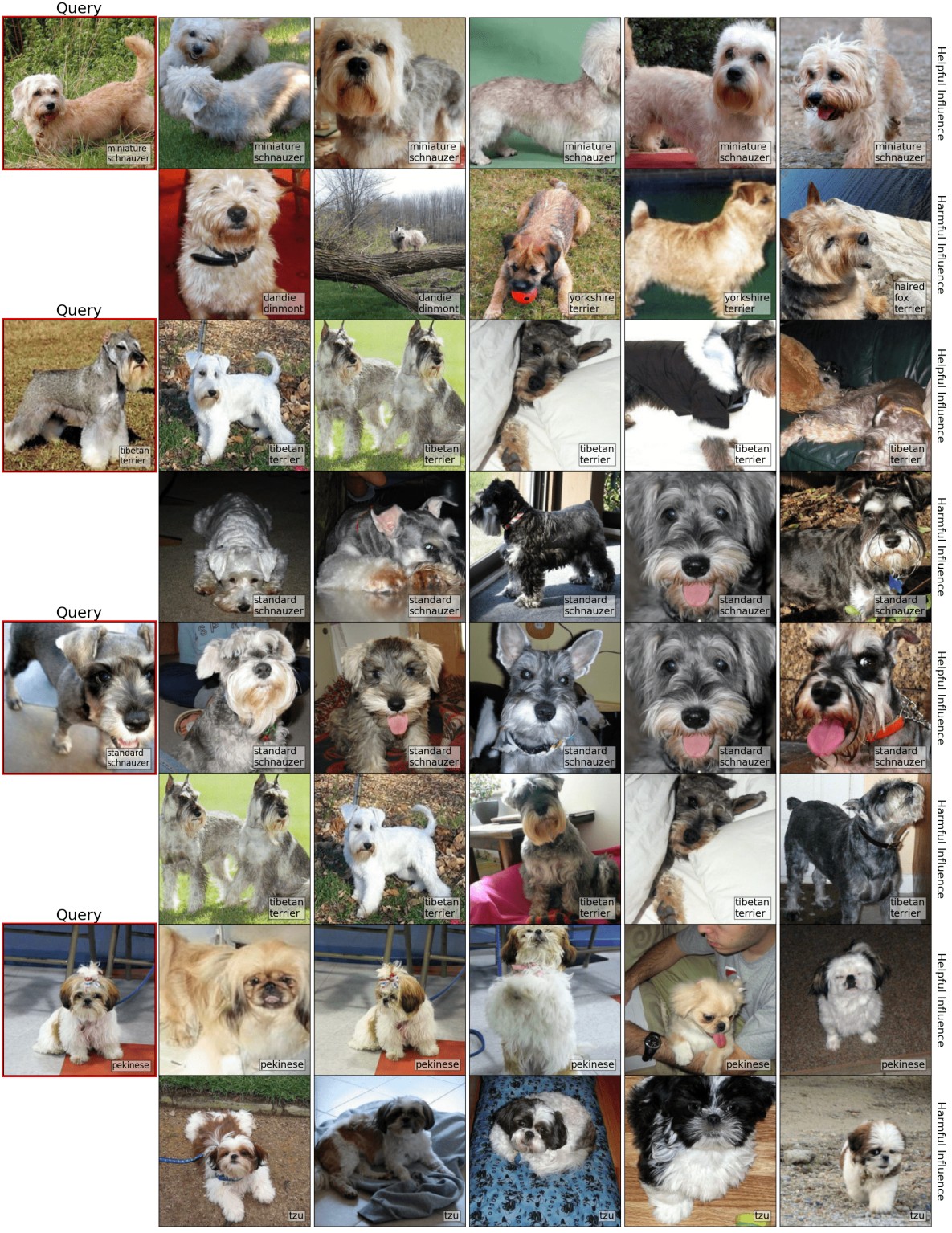

Figure 13: Query image, top helpful support images, and top harmful support images ranked via support influence for Dog dataset.

## A.8 Ranking FC Images By Euclidean Distance in Feature Space

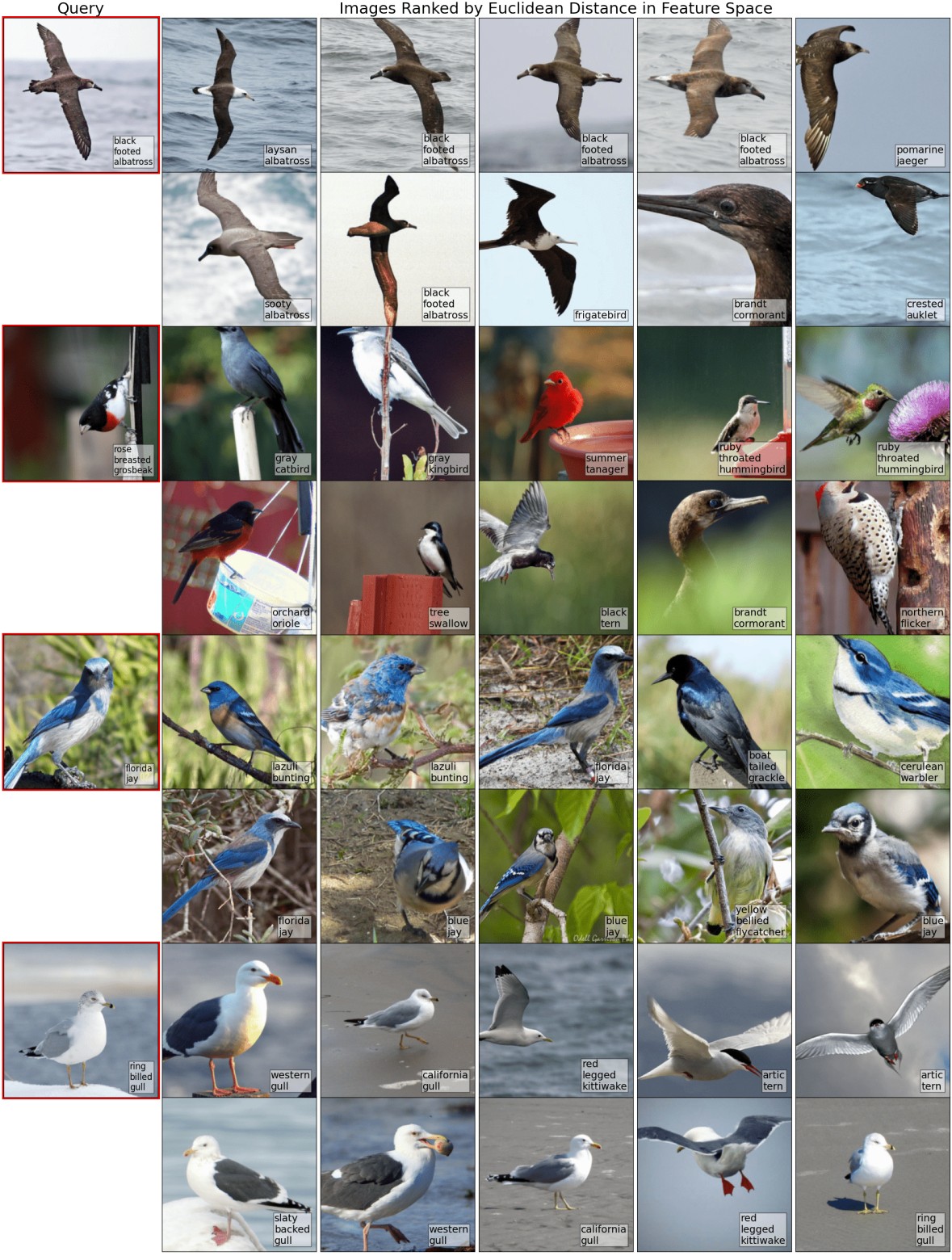

Figure 14: Query image and top images ranked by Euclidean distance in feature space of FC model for Bird dataset. The degree of semantic similarity between the query and the top images is lower than the NW model (see differences in class labels).

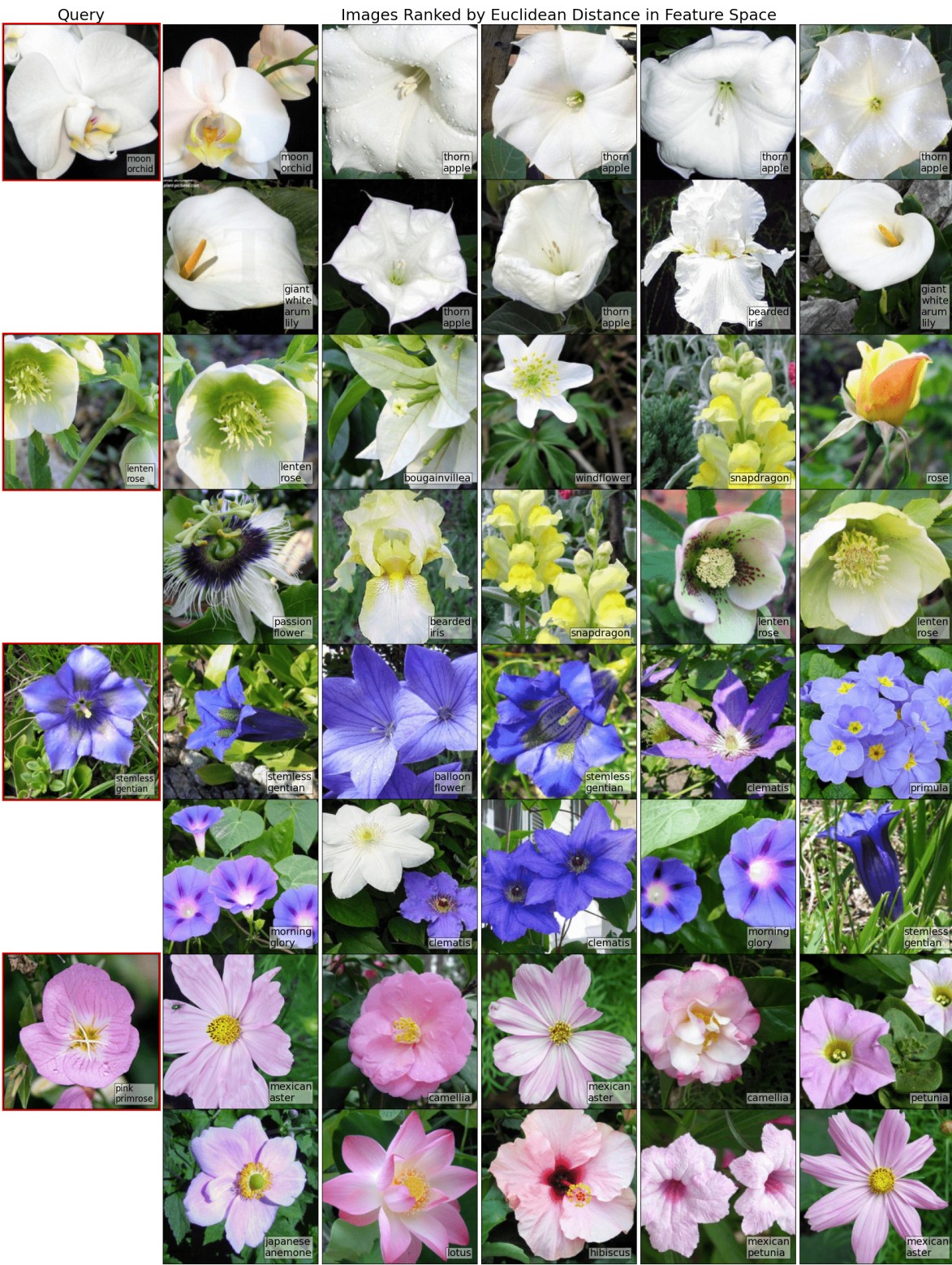

Figure 15: Query image and top images ranked by Euclidean distance in feature space of FC model for Flower dataset. The degree of semantic similarity between the query and the top images is lower than the NW model (see differences in class labels).

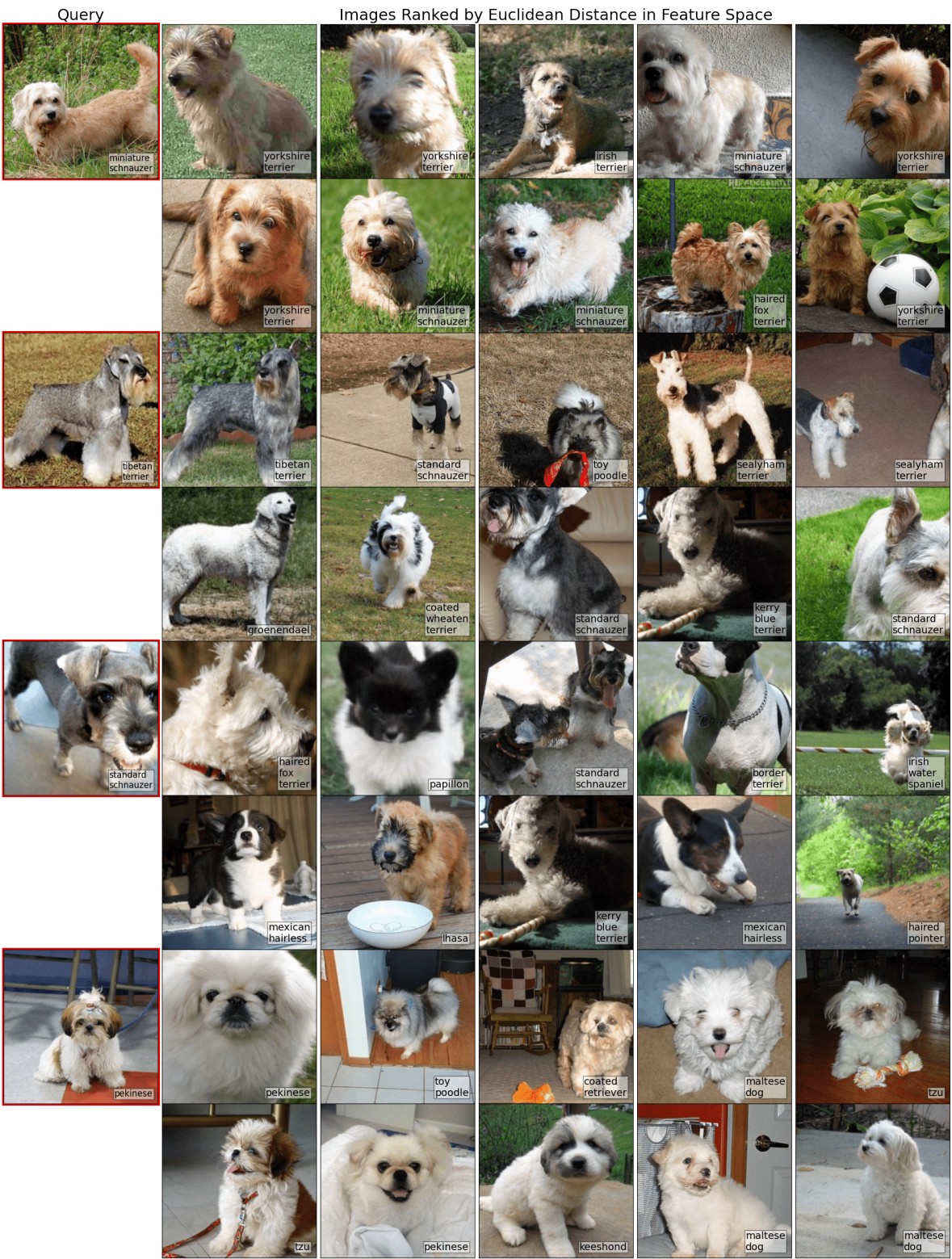

Figure 16: Query image and top images ranked by Euclidean distance in feature space of FC model for Dog dataset. The degree of semantic similarity between the query and the top images is lower than the NW model (see differences in class labels).

## A.9   Performance vs. support set size on ResNet-18

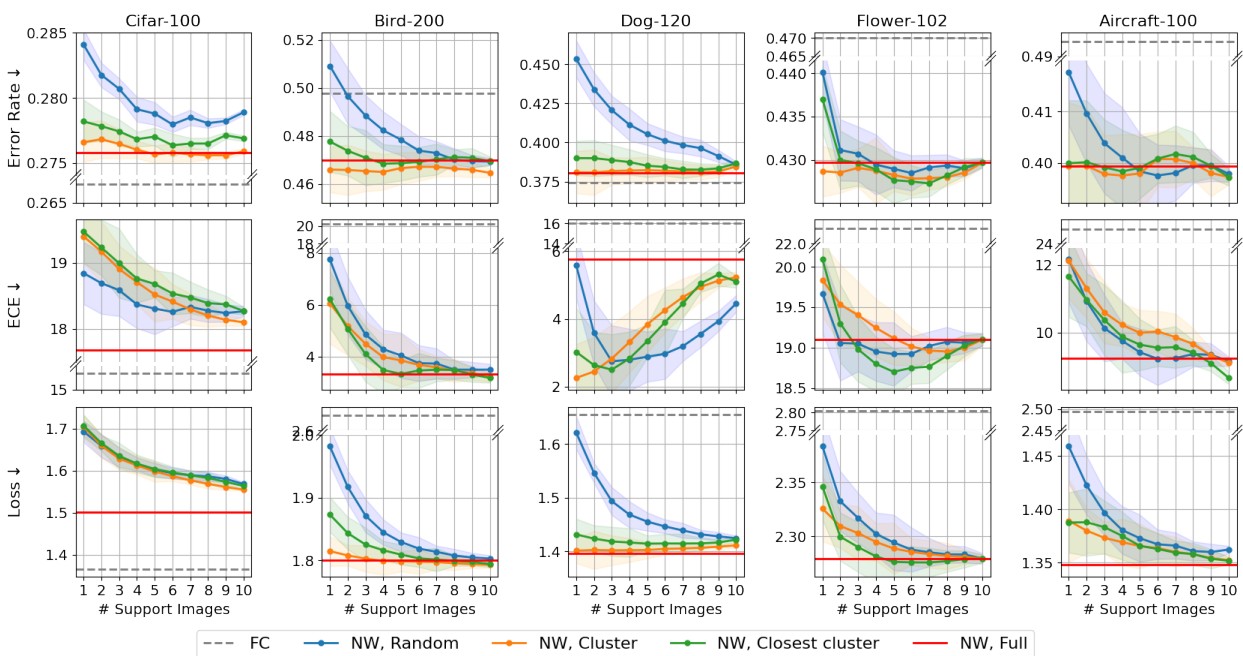

## A.10   Reliability diagram on DenseNet-121

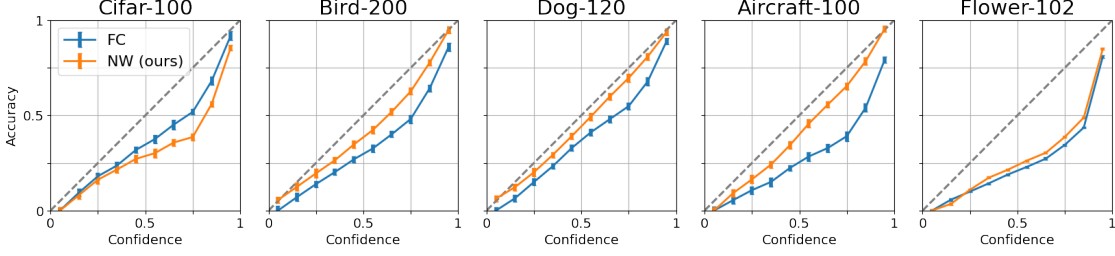

