# OpenReview forum: "A Flexible Nadaraya-Watson Head Can Offer Explainable and Calibrated Classification"
_TMLR — Accepted by TMLR_

### Review · Reviewer_Su24 · 2022-12-20

**Summary Of Contributions:**

In this paper, the authors proposed a Nadaraya-Watson (NW) classification head that can be attached to any deep convolutional feature extractor for image classification. In particular, the proposed NW head compares an input image ("query") with a set of support images in the feature space, and classifies the input image into a class whose support images have the highest total similarity with the input image in the feature space. A model with an NW head can be trained using a cross-entropy loss.

During inference, a support set can be the entire training set, a random subset of the training set, the cluster centers (in the feature space), or the closest training images to the cluster centers (in the feature space). The authors also proposed the concept of support influence, which quantifies the change in the cross-entropy loss on a test instance when a support element is removed.

In their experiments, the authors trained ResNet-18- and DenseNet-121-based NW models on the CUB-200-2011, Stanford Dogs, Oxford Flowers, and FGVC-Aircraft datasets, and found that their NW models achieved comparable performance with the baseline models with a fully-connected last layer. They also studied the top support images to given query images, as well as support images that are most helpful and most harmful to query images (defined by the highest/lowest support influences).


**Audience:**

No

**Broader Impact Concerns:**

There are no ethical concerns associated with this work.

**Claims And Evidence:**

Yes

**Requested Changes:**

Based on the above assessment, I recommend rejecting this paper. Since the idea is not novel, I encourage the authors to more carefully study the existing literature and to develop more novel ideas that can improve both the accuracy and interpretability over the current state-of-the-art.

**Strengths And Weaknesses:**

Strengths:
- The proposed NW head is simple and easy to implement and integrate into existing convolutional neural networks.
- The paper is clearly written and easy to follow.

Weaknesses:
- The idea is not novel. In fact, there have been more sophisticated deep prototype-based models that learn prototypical examples/prototypical parts of various image classes, and classify an input image by comparing the image with those prototypical examples/prototypical parts in the feature space (e.g., ProtoPNet by Chen et al., NeurIPS 2019). Those learned prototypical examples/prototypical parts can be understood as "support images." There are also deep k-NN based models that classify an input image by comparing the image with (patches of) training images in the feature space (e.g., visual correspondence model by Nguyen et al., NeurIPS 2022). In this case, the entire training set can be understood as "support images." All these models classify an input image by comparing it with "support images" in the feature space.
- The accuracy of the NW models (and also the baseline models) is low. Did you use ImageNet-pretrained models?
- There is a lack of comparison with other state-of-the-art models in terms of accuracy, and there is a lack of comparison with other interpretable models in terms of interpretability.

---

> ### Author Response · Authors · 2023-01-03
> **Response to Reviewer Su24, 1/2**
>
> We thank Reviewer Su24 for helpful feedback on our work. We address many of Reviewer Su24’s suggestions in the general response above and respond to specific comments below.
>
> > The idea is not novel. In fact, there have been more sophisticated deep prototype-based models that learn prototypical examples/prototypical parts of various image classes, and classify an input image by comparing the image with those prototypical examples/prototypical parts in the feature space (e.g., ProtoPNet by Chen et al., NeurIPS 2019). Those learned prototypical examples/prototypical parts can be understood as "support images." There are also deep k-NN based models that classify an input image by comparing the image with (patches of) training images in the feature space (e.g., visual correspondence model by Nguyen et al., NeurIPS 2022). In this case, the entire training set can be understood as "support images." All these models classify an input image by comparing it with "support images" in the feature space.
>
> Thank you for your comment and pointing out relevant papers; we have added these references to our revised manuscript.
>
> To address your comment, we will first summarize the above 2 papers. ProtoPNet works by learning a set of $m$ prototypes, which represent parts of an image (e.g., head of a bird) in the feature space. The authors propose a prototype layer which explicitly computes an aggregated similarity score as a function of similarities between each prototype and every patch in the input image. These similarity scores are further passed through a final fully connected layer. The training of this model involves a 3-phase optimization scheme. The authors show that their prototype layer allows for interpretability by visualizing learned prototypes, while exhibiting a trade-off in performance due to reduced model expressivity. In our current context, the ProtoPNet can be interpreted as a parametric model, where the prototype layer explicitly enforces interpretability as an inductive bias.
>
> The visual correspondence (VC) paper works by leveraging the features in pretrained/finetuned models. In particular, for a given query image, they rank the training images using similarity metrics computed in the feature space, and then take the dominant class of the top-k images as the predicted class. In the current context, VC can be viewed as a post-hoc explainable classification method, in the sense that it uses features from a parametric model to design a classifier in a nonparametric fashion.
>
> We believe that our method is inherently different due to the nonparametric and end-to-end training nature; the Nadaraya-Watson prediction “builds” its output as a function of pairwise similarities with neighboring points in the feature space. In short, the NW prediction explicitly encodes how each neighbor directly affects the prediction, and its modeling capacity scales with the size of the support set. The features are optimized for pairwise similarities, as we train the model end-to-end.
>
> In contrast to VC, we do not utilize pretrained/finetuned features of FC models as a post-training explainability method via a k-NN style classifier. Instead, we train our features end-to-end with a NW (soft k-NN) style classifier. In fact, we perform a similar analysis in Figure 5 and Figures 13-15 in the Appendix, where we visualize the top images ranked by Euclidean distance in the feature space of the FC model for a given query image. We find that the semantic similarity is much less than the NW head in Figure 7, 9, and 11 (see the disagreement in class labels between query and top images). This is likely due to the fact that feature-wise similarity is explicitly enforced in the NW head, whereas the FC head only needs to learn features which can be separated by hyperplanes. There is no guarantee that distances in feature space track visual similarity in the FC head.
>
> In contrast to a parametric ProtoPNet, we do not learn the prototypes, nor do we encode interpretability explicitly in the model architecture. Perhaps as a result of this last fact, we actually observe better results over a baseline FC net, which is not the case for ProtoPNet. For example, the authors report a 3.5% decrease in accuracy on Bird-200 dataset with their ProtoPNet vs. a parametric baseline, whereas our results indicate a ~5% increase in accuracy on the same dataset (note that due to variations in pre-processing, accuracy values can slightly vary between publications).

---

> > ### Author Response · Authors · 2023-01-03
> > **Response to Reviewer Su24, 2/2**
> >
> > > The accuracy of the NW models (and also the baseline models) is low. Did you use ImageNet-pretrained models?
> >
> > Following prior work and in order to keep our results general and disentangled from improvements possibly stemming from pretrained weights, we use randomly initialized weights for all models. We note that the FC (i.e. baseline) results are consistent with performance achieved by prior works (see, for example, [1]).
> >
> > > There is a lack of comparison with other state-of-the-art models in terms of accuracy, and there is a lack of comparison with other interpretable models in terms of interpretability.
> >
> > We would like to emphasize that the overarching objective of our paper is an empirical analysis of an NW head as a simple, flexible, and nonparametric alternative to the FC head. It is well established that non-parametric techniques like the NW head can offer better interpretability than parametric models, which we further demonstrate in our experiments. Perhaps surprisingly, however, we also find that the NW head can achieve highly accurate and well-calibrated prediction results over a range of datasets and architecture choices. Importantly, we present various inference modes for the NW head, which showcases its flexibility and allows users to explore different tradeoffs, such as inference time versus accuracy. Note that, we do not engineer any aspects of the model to improve interpretability or accuracy. We simply replace the last FC layer with a simple NW head and empirically analyze the performance implications. We acknowledge that there are many engineered, state-of-the-art methods that aim to improve accuracy, interpretability and/or calibration in various scenarios. Some of these techniques, such as the temperature scaling method we consider in the paper, can be combined with the NW head. However, in general, we regard this literature as out of the scope of our paper.
> >
> >
> > [1] Yun et al, Regularizing Class-wise Predictions via Self-knowledge Distillation, CVPR 2020

---

### Review · Reviewer_5nr3 · 2022-12-20

**Summary Of Contributions:**

This paper addresses the problem of calibration in deep learning. Given the output on a query, one desires the probability of error to be reflected by that number. The proposed method attaches to a neural network like a 'head', makes non-parametric decisions for classification, and obtains the outcome 'score' from a similar non-parametric approach.

This method could be a major contribution if the experiments were run on recent and practical datasets. However, most datasets in this study are ten years old and of small size. For these experiments, no error bars or quantiles are reported. Moreover, the visualization studies are run with a method that uses 50% more compute than other methods.


**Audience:**

Yes

**Broader Impact Concerns:**

N.A.

**Claims And Evidence:**

Yes

**Requested Changes:**

Adjustments:
  * Either report Table 1 and Figure 4 with error bars/quantiles, or evaluate on larger datasets, e.g. ImageNet.

Minor points:
 * Table 1 is confusing to interpret, because accuracy is better when higher, and ECE is better when lower. Why not report error rate instead of accuracy? Then both metrics are better when lower (or report 1-ECE so both metrics are better when higher).

Open question that is unrelated to this review:
* Could a special case of the non-parametric classifier be proven to be a proper scoring rule? In that case, there might be a bound on the ECE.


**Strengths And Weaknesses:**

Strengths:

  * Non-parametric approaches for calibration have been overlooked in recent studies of neural network calibration.
  * The Taylor expansion of the influence function (equation 2) can be computed exactly for this new 'head.'
  * Classically, the main pain point of non-parametric approaches is 'how to find the support set?' and 'what size support set to store?'. The paper addresses these questions finely with the comparisons of support sets in section 3.3.


Weaknesses:

 * The main weakness of this paper is the experimental evaluation. All datasets are roughly ten years old or older (2008, 2011, 2013) and do not reflect modern problems. If I counted correctly, all datasets are smaller than 15k images (CUB (6k), S Dogs (12k), Flowers (1k), Aircraft (10k)). Recent works show that larger datasets and larger models already yield 'good calibration' [1].
* --> It should be noted that small experimental results need not be a negative point in this review. However, suppose one chooses to use small datasets and small models, I expect better experimental conduct. For example, it seems quite easy to calculate error bars or quantiles for the results in Table 1. Also, the diagrams in Figure 4 could easily be plotted with error bars or quantiles for such small models and datasets.
* A small comment on Table 1, all major studies on neural network calibration have recommended tempering the predictions[1, 2], which has not been used here?
 * This new 'head' will require more memory and compute compared to fully connected layers. And on this note: Table 1 shows that 'NW, full' is best or second best on almost all datasets, and Table 3 shows that it's 50% more expensive in terms of compute. What applications do the authors have in mind for this method that takes so much more compute?
Likewise, all visualization studies in section 4.3 are made with the 'full support set', which is 50% more expensive in terms of compute. What are the conclusions from such visualization study when the model is compared with another model that uses the same amount of compute?

[1] Minderer et al. "Revisiting the calibration of modern neural networks." NeurIPS 2021
[2] Guo et al. "On calibration of modern neural networks." ICML 2017

---

> ### Author Response · Authors · 2023-01-03
> **Response to Reviewer 5nr3**
>
> We thank Reviewer 5nr3 for positive comments and helpful feedback on our work. We address many of Reviewer 5nr3’s suggestions in the general response above and respond to specific comments below.
>
> > The main weakness of this paper is the experimental evaluation. All datasets are roughly ten years old or older (2008, 2011, 2013) and do not reflect modern problems. If I counted correctly, all datasets are smaller than 15k images (CUB (6k), S Dogs (12k), Flowers (1k), Aircraft (10k)). Recent works show that larger datasets and larger models already yield 'good calibration' [1].
> --> It should be noted that small experimental results need not be a negative point in this review. However, suppose one chooses to use small datasets and small models, I expect better experimental conduct. For example, it seems quite easy to calculate error bars or quantiles for the results in Table 1. Also, the diagrams in Figure 4 could easily be plotted with error bars or quantiles for such small models and datasets.
>
> Thank you for the suggestion. We have added error bars to Table 1 in the old manuscript (now separated into Table 1 and Table 2 in the updated manuscript) and Figure 4.
>
> Additionally, we have added results on Cifar-100 (50k training examples with 100 classes) and iNaturalist-10k (500k training examples with 10k classes) datasets. We present Cifar-100 results for ResNet-18 and DenseNet-121 backbone feature extractors to Table 1 and Table 2 (note that we split Table 1 in the old manuscript to two tables, one for error rate and one for ECE). Following prior work [2], we present iNaturalist-10k on ResNet-50 feature extractor.
>
> We conclude from these additional results that the NW head exhibits generally comparable performance as compared to the FC head in terms of accuracy and calibration. The NW head pulls ahead in performance particularly for fine-grained classification tasks, and pulls ahead even further for small-medium sized datasets. We thank the reviewers for these suggestions - we have updated the manuscript wording to reflect these findings.
>
> > A small comment on Table 1, all major studies on neural network calibration have recommended tempering the predictions[1, 2], which has not been used here?
>
> In Table 2 of the old manuscript (now Table 3 in the updated manuscript), we show results on both FC and NW models with temperature scaling, as well as label smoothing. In particular, FC-TS and NW-TS are temperature scaling results.
>
> In general, we find that our method outperforms the FC method under both LS and TS techniques for the tasks we consider. In particular, the post-training technique of TS is a simple strategy that further improves our calibration performance.
>
> We would like to stress that our method exhibits improved calibration “out-of-the-box”, i.e. with no additional techniques to improve calibration; that is, we do not specifically target calibration in our architecture or training procedure. In particular, many effective calibration techniques proposed in the literature can be applied to the NW head, which we strive to highlight in Section 4.2.1.
>
> > This new 'head' will require more memory and compute compared to fully connected layers. And on this note: Table 1 shows that 'NW, full' is best or second best on almost all datasets, and Table 3 shows that it's 50% more expensive in terms of compute. What applications do the authors have in mind for this method that takes so much more compute? Likewise, all visualization studies in section 4.3 are made with the 'full support set', which is 50% more expensive in terms of compute. What are the conclusions from such visualization study when the model is compared with another model that uses the same amount of compute?
>
> We agree with the reviewer that ‘Full’ is computationally expensive as compared to FC and ‘Cluster’. Nevertheless, we believe that certain scenarios may warrant the added computational cost given the potential benefits. For example, in high-risk, safety-critical domains like medical imaging, high inference throughput is not as important as performance, interpretability, and explainability.
>
> In addition, we would like to clarify that these inference modes can be used interchangeably and flexibly. As an example, consider a workflow which would involve using Cluster mode to perform efficient inference, and then using Full mode on a select few (potentially problematic) test queries to understand model behavior. We stress that this flexibility at inference time is not possible with FC baseline models.
>
>
> > Table 1 is confusing to interpret, because accuracy is better when higher, and ECE is better when lower. Why not report error rate instead of accuracy? Then both metrics are better when lower (or report 1-ECE so both metrics are better when higher).
>
> Thank you for this suggestion. We have changed accuracies to error rates.

---

### Review · Reviewer_dLd3 · 2022-12-28

**Summary Of Contributions:**

The paper is presenting a new output layer for deep networks in the context of supervised learning. The proposed layer is a non-parametric model that achieves classification of given query instances regarding other support instances associated with classes, relying on distances in the feature space obtained with the deep network. Although such a model is not of outstanding originality – the authors are making a good overview of other works proposing similar non-parametric approaches for classification with deep networks – it makes the proposal very clear and well developed. In particular, the authors argue that such an approach allows some form of (partial) interpretation, similarly to attention mechanisms according to the classification weights of the supports, while achieving better calibrated output. Results are presented on some datasets to support the claims.

**Audience:**

Yes

**Broader Impact Concerns:**

No broader impact concerns with this paper.

**Claims And Evidence:**

No

**Requested Changes:**

Experiments with other datasets, especially larger datasets, should be conducted to validate that the approach proposed is not good only with the type of datasets used for evaluation (small-mid size image classification datasets with 100s-200s classes), and can work in other settings.

Full details on datasets (size, number of features, number of classes, other characteristics) and full details experimental setting should be provided. This can be in the appendix. It should be made to both apprehend better the results reported – I need to check on the Web for getting the details on the datasets and figure out how similar they were – and allow reproducibility of the results.

More explanations on the choice of hyperparameters used, showing that the choices for the values used have been properly studied, with possibly some ablation studies over some design choices, to validate that they are the right ones.

Experiments on calibration are not fair on some aspects, as the approach is already doing some form of calibration through the $\tau$ parameter of Eq. 5. Adjustment of this parameter is not detailed, but I am quite sure adjusting it would explain a big share of the proper calibration obtained. I think this part of the results should be revised completely to allow a better comparison with FC.


**Strengths And Weaknesses:**

Although it is not a very original topic, the paper achieves to show very clearly and in a relatively convincing manner the benefit and relevance of a non-parametric classification layer. I found the explanations provided to be quite clear, the level of details is appropriate (but the proposed approach is relatively simple by itself, which is good), and the relation to other related work is well made and does a proper overview (although a little short) of other related approaches.

I think this paper can have an impact in establishing a way for implementing a non-parametric classification output layer, which is of interest for a relatively broad part of the deep learning community. It can reveal to have a good citation level in a few years from now.

Regarding the weaknesses of the paper, I think it falls short in terms of experiments. It is currently limited to image classification over four datasets (bird, dog, flower, aircraft), which appear of similar limited size (10000s to 20000s instances) and similar number of classes (100s-200s classes per dataset). That’s quite a restrictive setting and the capacity of the approach to properly perform on different datasets, especially larger ones, is not clear to me. I would in fact guess that more classical models with fully connected output layers will behave much better with larger training sets and possibly outperform the proposed approach.

Another issue is the scaling of computation with the proposed approach, with larger datasets. Results in Table 3 show a slowdown by a factor of 50% with the full approach on the bird dataset, which is only of ~5000 training instances. I think performance may be hurt much more with larger datasets, as the algorithm will not scale very well, having an O(n^2) complexity regarding the dataset size, if implemented relatively naively.

Also, I note that the calibration results appear quite good in Table 1 when compared to FC output layer results, but are much closer when calibration with temperature scaling is done in Table 2. In fact, I consider that the temperature constant used for the proposed NW approach presented in Eq. 5 (temperature is the $\tau$) already leads to some form of temperature scaling for the proposed NW approach. Given this, I consider the evaluation of NW with non-calibration FC is somehow not fair and should not be presented in Table 1 and Figure 4. Even more, I am making the hypothesis that the good results in Table 2 results from a much finer calibration done with NW (a double calibration in fact, one in Eq. 5 and a subsequent one with temperature scaling).

Experiments settings are not well detailed on several aspects and I am not convinced that they can easily be reproduced only from the paper. Details given in page 6 are not enough, in my opinion. Also, given the lack of details on the hyperparameters used, it gets difficult to spot if any of them have great influence on the performances. More analysis and possibly some ablation studies would allow to increase confidence that the configuration used for the proposed approach is well optimized.

Arguments on explainability are mostly based on examples. That remains relatively weak, in my opinion, I would have liked to get stronger arguments than that. One may argue that the examples were cherry-picked and the general behavior may not be as nice as what is presented.

---

> ### Author Response · Authors · 2023-01-03
> **Response to Reviewer dLd3, 1/2**
>
> We thank Reviewer dLd3 for positive comments and helpful feedback on our work. We address many of Reviewer dLd3’s suggestions in the general response above and respond to specific comments below.
>
> > Regarding the weaknesses of the paper, I think it falls short in terms of experiments. It is currently limited to image classification over four datasets (bird, dog, flower, aircraft), which appear of similar limited size (10000s to 20000s instances) and similar number of classes (100s-200s classes per dataset). That’s quite a restrictive setting and the capacity of the approach to properly perform on different datasets, especially larger ones, is not clear to me. I would in fact guess that more classical models with fully connected output layers will behave much better with larger training sets and possibly outperform the proposed approach.
> Experiments with other datasets, especially larger datasets, should be conducted to validate that the approach proposed is not good only with the type of datasets used for evaluation (small-mid size image classification datasets with 100s-200s classes), and can work in other settings.
>
> Thank you for this suggestion. Prompted by this, we have added results on Cifar-100 (50k training examples with 100 classes) and iNaturalist-10k (500k training examples with 10k classes) datasets. We present Cifar-100 results for ResNet-18 and DenseNet-121 backbone feature extractors to Table 1 and Table 2 (note that we split Table 1 in the old manuscript to two tables, one for error rate and one for ECE). We present iNaturalist-10k on ResNet-50 feature extractor.
>
> We conclude from these additional results that the NW head exhibits generally comparable performance relative to the FC head in terms of accuracy and calibration. The NW head pulls ahead in performance particularly for fine-grained classification tasks, and pulls ahead even further for small-medium sized datasets. We thank the reviewers for these suggestions - we have updated the manuscript wording to reflect these findings, as well as elaborating on these findings in the added Discussion section.
>
> > Another issue is the scaling of computation with the proposed approach, with larger datasets. Results in Table 3 show a slowdown by a factor of 50% with the full approach on the bird dataset, which is only of ~5000 training instances. I think performance may be hurt much more with larger datasets, as the algorithm will not scale very well, having an O(n^2) complexity regarding the dataset size, if implemented relatively naively.
>
> We agree with the reviewer that computation is a limitation with the NW head (and, arguably, most nonparametric approaches) due to the nature of making comparisons between images.  Nevertheless, we believe that certain scenarios may warrant the added computational cost given the potential benefits. For example, in high-risk, safety-critical domains like medical imaging, high inference throughput is not as important as performance, interpretability, and explainability.
>
> In addition, we would like to reiterate that we explore several inference modes which each have advantages and trade-offs. In particular, we find that Cluster mode can be a sufficient replacement to Full mode (Figure 2), while being computationally cheaper (Table 2). These inference modes can be used interchangeably and flexibly. As an example, consider a workflow which would involve using Cluster mode to perform efficient inference, and then using Full mode on a select few (potentially problematic) test queries to understand model behavior. We stress that this flexibility at inference time is not possible with FC baseline models.

---

> > ### Author Response · Authors · 2023-01-03
> > **Response to Reviewer dLd3, 2/2**
> >
> > > Also, I note that the calibration results appear quite good in Table 1 when compared to FC output layer results, but are much closer when calibration with temperature scaling is done in Table 2. In fact, I consider that the temperature constant used for the proposed NW approach presented in Eq. 5 (temperature is the ) already leads to some form of temperature scaling for the proposed NW approach. Given this, I consider the evaluation of NW with non-calibration FC is somehow not fair and should not be presented in Table 1 and Figure 4. Even more, I am making the hypothesis that the good results in Table 2 results from a much finer calibration done with NW (a double calibration in fact, one in Eq. 5 and a subsequent one with temperature scaling).
> > Experiments on calibration are not fair on some aspects, as the approach is already doing some form of calibration through the  parameter of Eq. 5. Adjustment of this parameter is not detailed, but I am quite sure adjusting it would explain a big share of the proper calibration obtained. I think this part of the results should be revised completely to allow a better comparison with FC.
> >
> > We would like to clarify that for all experiments except for those in Section 4.2.1 (Improving Calibration), we do not adjust the temperature in Eq. 5 and set $\tau=1$ (see “Network Architecture and Hyperparameters” subsection). Thus, we believe that the results in Table 1 and Figure 4 are fair in the sense that no temperature scaling was performed for either head. Furthermore, all temperature scaling experiments in Section 4.2.1 were performed identically between FC and NW heads to produce Table 2 in the old manuscript (now Table 3 in the updated manuscript). We apologize for the confusion - we have added Appendix A.3 to clarify this in the revised manuscript.
> >
> > > Experiments settings are not well detailed on several aspects and I am not convinced that they can easily be reproduced only from the paper. Details given in page 6 are not enough, in my opinion. Also, given the lack of details on the hyperparameters used, it gets difficult to spot if any of them have great influence on the performances. More analysis and possibly some ablation studies would allow to increase confidence that the configuration used for the proposed approach is well optimized.
> > More explanations on the choice of hyperparameters used, showing that the choices for the values used have been properly studied, with possibly some ablation studies over some design choices, to validate that they are the right ones.
> > Full details on datasets (size, number of features, number of classes, other characteristics) and full details experimental setting should be provided. This can be in the appendix. It should be made to both apprehend better the results reported – I need to check on the Web for getting the details on the datasets and figure out how similar they were – and allow reproducibility of the results.
> >
> > Thank you for this suggestion. In response to this comment, we have added detailed experimental results and dataset information to Sections A.1 and A.2 in the appendix. We have also presented results on different settings of the hyperparameters N_s (training support set size) and N_b (batch size) in A.6.
> >
> > > Arguments on explainability are mostly based on examples. That remains relatively weak, in my opinion, I would have liked to get stronger arguments than that. One may argue that the examples were cherry-picked and the general behavior may not be as nice as what is presented.
> >
> > Thank you for this suggestion. We have added Figure 5 to the updated manuscript, which shows the percentage of top support set labels which match the query label. Top labels are found by ranking support images by negative Euclidean distance and taking the top 10 images. The NW model has a higher degree of semantic similarity with its top support images, which corroborates our claims previously made through visual arguments.

---

> > > ### Comment · Reviewer_dLd3 · 2023-01-17
> > > **Paper update**
> > >
> > > I read authors' answers and all comments. The authors worked diligently in updating the paper following the reviewers' comments and the result is convincing, in my opinion. Although it is not a highly original paper, it brings together several ideas, proposes and analyze an output layer that makes sense. With the new results on larger datasets, the analysis is better nuanced and reflect better the context where the proposed NW layer is effective. I think the paper would be of interest for the community.

---

### Author Response · Authors · 2023-01-03
**Response to all reviewers**

We thank the reviewers for their thoughtful and constructive review of our manuscript. In response to the feedback, we provide a general response here to points raised by multiple reviewers, and provide reviewer-specific, point-by-point responses below.

We emphasize that we wish to position the NW head as a simple, easy-to-implement, nonparametric alternative to the FC head. To the best of our knowledge, we believe this is the first work to explore such a head in the context of modern convolutional architectures. Despite its simplicity, in the tasks that we consider, we find that the benefits of this alternative include:

+ More interpretable and better calibrated predictions.
+ More flexibility for practitioners, due to allowing for multiple inference “modes'' corresponding to different choices of the support set at test-time.
+ Precise explainability via an easy-to-compute influence function. Note that prior works on applying influence functions to deep networks for explainability [1] require approximations and cannot be computed in closed form.

In response to reviewers dLd3 and 5nr3 regarding lack of evaluation on large datasets, we have added results on Cifar-100 (50k training examples with 100 classes) and iNaturalist-10k (500k training examples with 10k classes) datasets. We present Cifar-100 results for ResNet-18 and DenseNet-121 backbone feature extractors in Table 1 and Table 2 (note that we split Table 1 in the old manuscript into two tables, one for the error rate and one for ECE). Following prior work [2], we present iNaturalist-10k on the ResNet-50 feature extractor.

We conclude from these additional results that the NW head exhibits generally comparable performance relative to the FC head in terms of accuracy and calibration. The NW head pulls ahead in performance particularly for fine-grained classification tasks, and pulls ahead even further for small-medium sized datasets. We thank the reviewers for these suggestions - we have updated the manuscript wording to reflect these findings, as well as elaborating on these findings in the added Discussion section.

Other minor changes:
+ Changing accuracies to error rates, so that lower is better for all metrics
+ Adding error bars (variances over 3 runs of each model/experiment) to figures and tables.
+ Adding details on experimental setup, including datasets, model details, calibration experiments, and hyperparameter tuning to the Appendix.

[1] Koh et al. Understanding black-box predictions via influence functions, 2017.
[2] Zhou et al. BBN: Bilateral-Branch Network with Cumulative Learning for Long-Tailed Visual Recognition, 2020

---

### Decision · Action_Editors · 2023-02-08

**Recommendation:** Accept as is

**Comment:**

The submission investigates a nonparametric alternative to the fully-connected output layer for deep neural network classifiers. The Nadaraya-Watson (NW) prediction head, as it is called, makes a prediction for a query example in the form of a weighted average of the support set labels. The weights themselves are expressed as a function of the Euclidean distance (in the space of embeddings output by the neural network) between the query example and individual support examples, and are normalized to sum up to 1.

The NW prediction head's properties are analyzed empirically through experiments on CUB-200-2011, Stanford Dogs, Oxford Flowers, and FGVC-Aircraft and are compared against those of a conventional fully-connected prediction head.

Reviewers find the proposal very clear and well developed (dLd3, Su24), and relevant to the community (dLd3), especially given that non-parametric approaches for calibration have been overlooked in recent studies on neural network calibration (5nr3). Concerns on consistency of the presentation in the tables (5nr3), error bars (5nr3), details of the experimental setup (dLd3), and missing connections to related works on deep prototype-based models (Su24) are adequately addressed by the authors in their response.

The main outstanding reviewer concerns are:

1. The submission's experiments are limited to image classification with relatively old datasets of similar limited size and similar numbers of classes (dLd3, 5nr3). In the context of recent works showing that larger datasets and larger models already yield "good calibration", Reviewer 5nr3 would like to see experiments at a larger scale to determine whether the NW head's calibration benefits are still significant in that regime.
2. The NW head has poor scaling properties with respect to dataset size (dLd3, 5nr3).

In their response, the authors present results on two new, larger datasets (CIFAR-100, iNaturalist-10k). They acknowledge the computational shortcomings of the NW head, but point to the Cluster mode as a cheap and effective replacement for the Full mode. They also argue that high-risk, safety-critical domains like medical imaging put less emphasis on high inference throughput and more emphasis on performance, interpretability, and explainability.

Opinions remain split on acceptance. While it is true that investigating even larger data regimes would strengthen the submission, the submission as it stands is well-supported given the scope of the claims made, and reviewers agree that the topic is of interest to the TMLR community.

I therefore recommend acceptance.

**Audience:**

Reviewers agree that the investigated approach is of interest to the TMLR audience.

**Claims And Evidence:**

The claims made in the submission are supported by accurate, convincing, and clear evidence for the data regimes investigated.